# Collab-RAG: Boosting Retrieval-Augmented Generation for Complex Question Answering via White-Box and Black-Box LLM Collaboration

**Ran Xu**[1*]**, Wenqi Shi**[2*]**, Yuchen Zhuang**[3]**, Yue Yu**[3]**, Joyce C. Ho**[1]**, Haoyu Wang**[4]**, Carl Yang**[1]

[1] Department of Computer Science, Emory University
[2] University of Texas Southwestern Medical Center
[3] College of Computing, Georgia Institute of Technology
[4] Department of Computer Science, SUNY Albany
`{ran.xu, j.carlyang}@emory.edu`

## Abstract

Retrieval-Augmented Generation (RAG) systems often struggle to handle multi-hop question-answering tasks accurately due to irrelevant context retrieval and limited complex reasoning capabilities. We introduce `Collab-RAG`, a collaborative training framework that leverages mutual enhancement between a white-box small language model (SLM) and a black-box large language model (LLM) for RAG. Specifically, the SLM decomposes complex queries into simpler sub-questions, thus enhancing the accuracy of the retrieval and facilitating more effective reasoning by the black-box LLM. Concurrently, the black-box LLM provides feedback signals to improve the SLM's decomposition capability. We observe that `Collab-RAG` relies solely on supervision from an affordable black-box LLM without additional distillation from frontier LLMs, yet demonstrates strong generalization across multiple black-box LLMs. Experimental evaluations across five multi-hop QA datasets demonstrate that `Collab-RAG` substantially outperforms existing black-box-only and SLM fine-tuning baselines by 1.8%-14.2% on average. In particular, our fine-tuned 3B SLM surpasses a frozen 32B LLM in question decomposition, highlighting the efficiency of `Collab-RAG` in improving reasoning and retrieval for complex questions. Our implementation is available at https://github.com/ritaranx/Collab-RAG/.

## 1 Introduction

Despite the strong performance of Large Language Models (LLMs) across a wide range of language tasks, they face several limitations such as hallucinations (Shi et al., 2024a), and difficulties adapting to evolving or domain-specific knowledge (Zhang et al., 2024b). Retrieval-Augmented Generation (RAG) has emerged as a powerful technique to address these challenges by integrating external knowledge sources, enabling LLMs to improve factual accuracy (Lewis et al., 2020) and response reliability (Lin et al., 2024) to their responses.

In a standard RAG pipeline, a retriever first searches from external corpora for relevant information based on a given query. The retrieved context is then combined with the query and fed into the LLM, allowing it to generate a more contextually grounded response. Although stronger black-box LLMs have shown strong capability in language modeling and have been used as RAG readers in practice (Shi et al., 2024a; Jeong et al., 2024), the standard RAG pipeline often has low-recall issues, where relevant information cannot be retrieved (Yu et al., 2024a). This issue is particularly pronounced in complex question-answering (QA) tasks, where multiple pieces of evidence are required for accurate reasoning.

To enhance the ability of black-box LLMs for handling complex questions in RAG applications, several studies have attempted to improve retrieval quality, such as training an

---

*Equal Contribution.

auxiliary model for text embedding (Shi et al., 2024a) or query re-ranking (Mao et al., 2024). However, these approaches primarily focus on refining single-step retrieval and fail to address the inherent challenges of complex question-answering scenarios, where iterative evidence gathering is essential for assembling the most relevant information from the corpus. Moreover, Jiang et al. (2023); Li et al. (2025); Liu et al. (2024a); Wang et al. (2024) propose training-free methods to leverage LLM itself to refine queries or perform multi-turn retrieval. However, without dedicated training, LLMs have limited capabilities in effective query decomposition and refinement, making these methods suboptimal for complex retrieval tasks. Recently, some studies finetune small language models to improve RAG performance (Liu et al., 2024b; 2025; Wei et al., 2025), however, updating parameters for black-box LLMs remains inefficient and resource-intensive, limiting the practicality of these methods in real-world applications. To summarize, it is still crucial yet challenging to fully unleash the capability of black-box LLMs for complex question answering tasks.

In this work, we introduce Collab-RAG, an RAG framework that enhances black-box LLMs for complex question answering by incorporating an additional white-box small language model (SLM). As shown in Figure 1, the SLM serves as a *decomposer*, breaking down complex queries into smaller, atomic subquestions to improve the retrieval of relevant contexts (Patel et al., 2022; Khot et al., 2023). The black-box LLM then acts as a *reader*, generating intermediate answers for each subquestion and synthesizing them into a final response. By structuring retrieval around decomposed subquestions, this framework effectively harnesses the context extraction capabilities of black-box LLMs to answer the complex questions step-by-step.

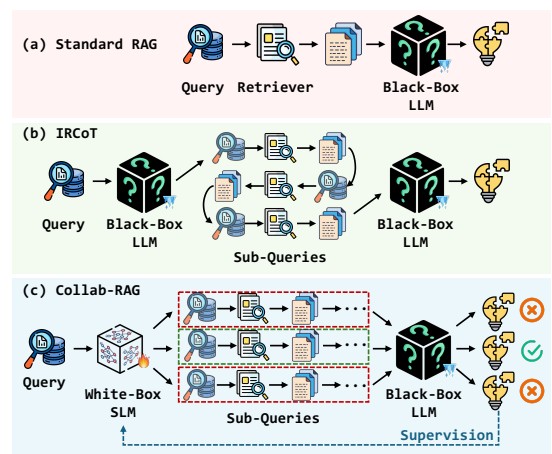

Figure 1: Comparison of various LLM-based RAG pipelines. Collab-RAG fosters collaboration between the SLM query decomposer and the LLM reader, allowing them to enhance each other.

Using SLMs directly for question decomposition is often ineffective, as they are not extensively trained on relevant data, and collecting large-scale, high-quality annotations is expensive. To overcome this challenge, we propose a *self-improving training strategy* that depends solely on feedback from black-box LLMs. During training, we model the black-box LLM (GPT-4o-mini) as an environment that generates responses alongside queries and retrieved contexts. The SLM engages in multi-turn interactions with this environment to iteratively refine its decomposition strategy. Our key insight is that *higher-quality question decomposition leads to better final answers*. Based on this, we design an iterative preference optimization approach (Pang et al., 2024) that uses feedback from the black-box LLM to improve the SLM's decomposition capabilities. Specifically, inspired by (Guo et al., 2025; Jin et al., 2025), we adopt a *rule-based* evaluation to distinguish between effective and ineffective decompositions by assessing both the format of the subquestions and the accuracy of the final answer. This preference optimization framework enables the SLM to learn optimal decomposition strategies that enhance retrieval and reasoning ability for the RAG pipeline, without relying on costly human annotations or distillation from frontier-class LLMs.

Our contributions can be summarized as follows: (i) *Problem Wise*, we propose Collab-RAG to enable dynamic collaboration between white-box SLMs and black-box LLMs, which have not yet been widely explored in existing RAG studies for complex question answering. (ii) *Methodology Wise*, Collab-RAG employs an iterative preference optimization approach using outcome feedback from black-box LLMs *without* relying on distillation from frontier LLMs. Collab-RAG is trained using feedback from GPT-4o-mini only but shows strong generalization across various LLMs. (iii) *Experimental Wise*, Collab-RAG outperforms both black-box LLM-only approaches and small LLM fine-tuning baselines by 1.8%-14.2%, demonstrating improved reasoning and retrieval effectiveness in complex question-answering tasks. No-

tably, for question decomposition, a fine-tuned 3B SLM achieves better results than a frozen 32B LLM on average, justifying the efficiency and effectiveness of `Collab-RAG`.

## 2 Related Work

**Retrieval-Augmented Generation.** RAG enhances LLMs by integrating external knowledge retrieval, improving the accuracy and relevance of generated responses. Earlier works study improving *retrievers* for RAG, as Shi et al. (2024a); Shao et al. (2023) finetune retrievers based on language model feedbacks. Then, with more available open-source LLMs, several works also design effective instruction finetuning pipelines towards RAG applications via collecting diverse training data (Lin et al., 2024; Liu et al., 2024b; Yu et al., 2024b), generating synthetic data (Xu et al., 2024; Zhu et al., 2024; Shi et al., 2024b), or incorporating chain-of-thought reasoning process (Yu et al., 2024a; Wei et al., 2025). More recently, reinforcement learning (RL) techniques have been employed to optimize retrieval relevance (Dong et al., 2024) and enhance the quality of chain-of-thought reasoning (Liu et al., 2025; Zhang et al., 2024a). RAG-Gym (Xiong et al., 2025) and RAG-Star (Jiang et al., 2024) employ reward models to guide LLM generation, but they rely on supervision from GPT-4 series models, which introduces additional supervision costs. Different from these works, we leverage a preference-based fine-tuning method to better decompose the complex questions based on the feedback of final answer, which alleviates the need for intermediate passage relevance supervision and can serve as a generic plug-in for LLMs.

**Query Optimization.** To improve end-to-end performance of RAG pipelines, several query optimization techniques have been proposed. Several studies have explored query rewriting techniques to enhance response quality (Ma et al., 2023; Mao et al., 2024), particularly in conversational question-answering systems where rewriting helps better capture user intent (Mo et al., 2023). More related to our work, some approaches focus on decomposing complex queries to facilitate step-by-step reasoning. Prompt-based decomposition methods have been proposed for reasoning tasks (Khot et al., 2023; Khattab et al., 2022) and have been extend to RAG scenarios (Verma et al., 2024; Liu et al., 2024a), while some other approaches refine queries progressively using retrieved contexts (Yu et al., 2023; Li et al., 2025). Additionally, recent methods leverage knowledge distillation from proprietary models to learn query decomposition strategies (Chan et al., 2024).

## 3 Preliminaries

**Retrieval-Augmented Complex Question Answering.** Standard open-domain QA typically relies on single-step retrieval to locate relevant information. In contrast, complex QA (also referred to as multi-hop QA), requires extracting and integrating *multiple pieces of information* while performing multi-step reasoning (Ho et al., 2020; Yang et al., 2018; Trivedi et al., 2022). In this work, we focus on enhancing RAG with LLMs for complex QA tasks, specifically targeting complex QA datasets. We do not study single-step QA in this work.

**Problem Formulation.** We consider a complex QA task that requires multi-step reasoning and the retrieval of external knowledge to derive solutions. Specifically, given a question $x_i \in \mathcal{X}$ that necessitates reasoning across multiple documents, the objective is to generate a comprehensive solution composed of (i) a decomposition of original question to multiple simpler sub-questions $\mathcal{Q}_i = \{q_{i,1}, q_{i,2}, \ldots, q_{i,T_i}\}$ with intermediate solutions $\mathcal{Y}_i = \{y_{i,1}, y_{i,2}, \ldots, y_{i,T_i}\}$ and (ii) a final answer $y_i \in \mathcal{Y}$, where $T_i \in \mathbb{N}^+$ denotes the number of reasoning steps. Each sub-question $q_{i,t}$ is used to iteratively retrieve relevant information from a corpus $\mathcal{C}$ via a retriever $r_\psi(\cdot)$ that progressively leads to the final answer.

## 4 Methodology

In this section, we introduce `Collab-RAG`, an advanced RAG framework designed for complex QA by leveraging collaboration between a white-box SLM and a black-box LLM. Denote $\mathcal{Q}^*$ as the set of finite sequences for subquestions and $\mathcal{Y}^*$ as the set of possible solutions, our framework `Collab-RAG` consists of two parts: (i) **a white-box small language model** $f_\theta(\cdot)$, to serve as a question decomposer. The decomposition function $f_\theta : \mathcal{X} \to \mathcal{Q}^*$ maps the

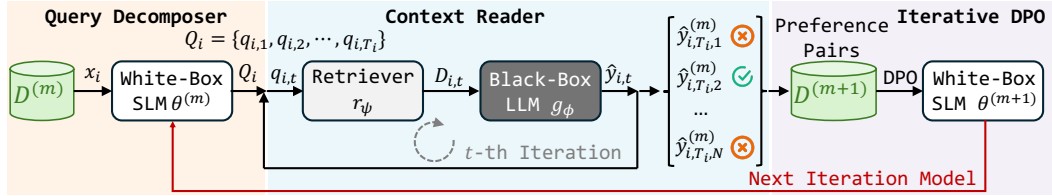

Figure 2: The iterative training framework of `Collab-RAG`. The SLM updates its parameters based on the generation quality of the LLM reader. The above process is conducted over multiple iterations to gradually improve SLM's decomposition capability.

question to a variable-length sequence of retrieval queries. (ii) **a black-box large language model** $g_\phi(\cdot) : \mathcal{Q} \times \mathcal{C} \to \mathcal{Y}^*$ to serve as the reader for answering the intermediate questions and deriving the final answer with retrieved contexts. Further details are provided below.

## 4.1 Overview of `Collab-RAG`

In our framework, the SLM is employed as a query decomposer to transform complex queries into simpler, manageable step-by-step subquestions (Section 4.3). Subsequently, the decomposed subquestions guide the LLM-based RAG system in sequentially retrieving and synthesizing relevant information. We view the RAG system as an environment (Section 4.2) and construct a diverse and extensive training dataset through interactions between the white-box SLM decomposer and the black-box LLM reader (Section 4.4). Since updating a black-box LLM is costly, we refine the system by updating the parameters of the SLM $f_\theta(\cdot)$. This is achieved through iterative preference optimization, leveraging feedback from the LLM (Section 4.5). The framework of `Collab-RAG` is illustrated in Figure 2.

## 4.2 RAG as Environment

We formulate the LLM-based RAG system as an environment that interacts with the LLM question decomposer. At each step $t \in \{1, 2, ..., T_i\}$, a retriever $r_\psi$ takes the current question $q_{i,t}$ and searches over a set of documents $D_{i,t} = r_\psi(q_{i,t})$ that are most relevant to $q_{i,t}$. Next, a generator $g_\phi$ uses the current question $q_{i,t}$, the retrieved knowledge $D_{i,t}$, and all previous responses $\{\hat{y}_{i,k}\}_{k=1}^{t-1}$ to produce a response $\hat{y}_{i,t}$ as:

$$\hat{y}_{i,t} = g_\phi(q_{i,t}, D_{i,t}, \{\hat{y}_{i,k}\}_{k=1}^{t-1}). \tag{1}$$

After the $T_i$ steps, the RAG system outputs the final solution $\hat{y}_{i,T_i}$. The quality of the question decomposition is evaluated using a reward function $u : \mathcal{X} \times \mathcal{Q}^* \to \mathbb{R}$ defined as a combination of format and accuracy rewards (Guo et al., 2025):

$$u(x_i, f_\theta(x_i)) = \begin{cases} \text{eval}(x_i, \hat{y}_{i,T_i}, y_i) & \text{if } \text{format}(x_i, \hat{y}_{i,T_i}) = 1; \\ 0 & \text{otherwise.} \end{cases} \tag{2}$$

Here, the **format reward** ($\text{format}(x_i, \hat{y}_{i,T_i})$) mainly check whether the model can output the decomposed question with correct reference of answers from previous round of answers[1], and the **accuracy reward** ($\text{eval}(\cdot)$) evaluates whether the response is correct, where $y_i$ is the ground-truth answer. In our study, we use a combination of *Exact Match* (EM) and *Accuracy* (Acc)[2] as the final accuracy reward, defined as $\text{eval}(x_i, \hat{y}_{i,T_i}, y_i) = 0.5 \times (\text{EM}(\hat{y}_{i,T_i} = y_i) + \text{Acc}(\hat{y}_{i,T_i} = y_i))$. A higher reward indicates a more effective decomposition $f_\theta(x)$ in guiding the RAG system to the correct answer.

---

[1]We enforce the model to output correct reference to previous round answers such as '*What is the name of the magazine mentioned in #1?*', where #1 is the answer for subquestion 1. However, sometimes the model will still output '*What is the name of the magazine mentioned in Question 1*', which yields suboptimal retrieval results.

[2]Accuracy only requires the ground truth answer exist in the response, which is relaxed from EM.

### 4.3 White-Box SLM $f_\theta$ as Query Decomposer

Complex questions frequently include multiple interdependent sub-questions, which standard retrieval systems often struggle to resolve effectively. To address this, we use a lightweight, white-box SLM specifically as a query decomposer. Given a complex query $x_i$, the decomposer $\theta$ generates a structured sequence of simpler sub-questions:

$$f_\theta(x_i) = \{q_{i,1}, q_{i,2}, \ldots, q_{i,T_i}\}, \tag{3}$$

where each sub-question $q_{i,t}$ clearly specifies the retrieval target, enhancing retrieval relevance and efficiency.

### 4.4 Black-Box LLM $g_\phi$ as Context Reader

Despite their effectiveness in text generation tasks, LLMs still struggle to systematically decompose and interpret complex questions in RAG systems due to: (1) limited multi-step reasoning for effective decomposition in smaller models, (2) suboptimal decompositions from zero-shot scenarios, and (3) difficulties integrating evidence dynamically across reasoning steps. To address these limitations, we formulate the RAG system as an interactive environment, utilizing feedback from the black-box LLM context reader as the supervision signal. Specifically, we regard a decomposition as positive if the black-box LLM can accurately interpret retrieved information and produce correct answers based on the decomposed queries, thereby improving the overall capability of the RAG system. For each input $x_i$, we initially sample sub-questions using the LLM-based decomposer: $Q_i = \{q_{i,t}\}_{t=1}^{T_i} \sim f_\theta(x_i)$. We then select the best-of-N decomposition based on the highest reward $u(x_i, f_\theta(x_i))$ as the positive example, and the worst-of-N decomposition as the negative example[3]:

$$\begin{cases} Q_{i+} = Q_{i,j}, \ j = \arg\max_{1 \leq n \leq N} u(x_i, Q_{i,n}), \\ Q_{i-} = Q_{i,j}, \ j = \arg\min_{1 \leq n \leq N} u(x_i, Q_{i,n}). \end{cases} \tag{4}$$

To prevent overfitting and encourage generalization, we follow reinforcement learning with human feedback (RLHF) principles (Ouyang et al., 2022) and ensure each positive and negative pair shares the same input prompt, thus constructing a balanced training dataset $\mathcal{D}_{\text{iDPO}} = \{(x_i, Q_{i+}, Q_{i-})\}$.

### 4.5 Iterative Preference Optimization via LLM Feedback

Directly employing SLMs for question decomposition often falls short due to their limited reasoning capabilities, while collecting extensive, high-quality annotations is resource-intensive. To overcome these limitations, we leverage feedback from black-box LLMs to curate training data for SLMs (Choi et al., 2024). It starts with supervised fine-tuning (SFT) based on rejection sampling and subsequently improves through iterative preference optimization guided by black-box LLM feedback.

**Warmup: Supervised Fine-tuning with Rejection Sampling.** Smaller language models often struggle with reinforcement learning due to their limited capabilities, leading to unstable training and poor convergence (Zheng et al., 2023). To address this challenge, we first fine-tune the model on self-generated high-quality data: For each question $x_i$, we employ rejection sampling (Zelikman et al., 2022) to generate candidate decompositions: $\{Q_{i,1}, Q_{i,2}, \cdots, Q_{i,N}\} \sim f_\theta(x_i)$. Then, only prompts with decomposition having the *highest reward* will be included in the SFT dataset $\mathcal{D}_{\text{SFT}} = \{(x_i, Q_i) \mid u(x_i, Q_i) \geq 0.5\}$. The SLM is then fine-tuned using next-token prediction loss, conditioned on the input question:

$$\mathcal{L}_{\text{SFT}} = -\mathbb{E}_{(x_i, Q_i) \sim \mathcal{D}_{\text{SFT}}} \left[ \sum_{l=1}^{L} \log f_\theta(Q_i[l] \mid Q_i[< l], x_i) \right], \tag{5}$$

---

[3]The prompt will be discarded if all decompositions lead to the same reward (e.g., all of them result in correct/incorrect answers).

where $Q_i[l]$ is the $l$-th token in the generated decomposition. Through SFT, the white-box SLM acquires the essential query decomposition skills needed for subsequent optimization.

**Iterative DPO.** Directly using decompositions generated by SLMs can lead to overfitting due to imbalances between positive and negative examples, resulting in simplistic patterns and limited model improvements. To mitigate this, we propose an iterative optimization framework that interleaves data collection and model training. Initially, we set the model parameters to $\theta^{(0)} = \theta$ after the warm-up phase and collect an initial dataset $\mathcal{D}^{(0)}$. At iteration $m$, we optimize model parameters $\theta^{(m)}$ and generate fresh samples $\{Q_{i,0}^{(m)}, Q_{i,1}^{(m)}, \cdots, Q_{i,K}^{(m)}\} \sim f_{\theta^{(m)}}(x)$. Following Section 4.4, we then construct preference pairs, forming the training dataset $\mathcal{D}_{\text{iDPO}}^{(m)} = (x_i, Q_{i+}^{(m)}, Q_{i-}^{(m)})$. To update the model for the next iteration $\theta^{(m+1)}$, we employ the direct preference optimization (DPO) for parameter optimization, using the model from the previous iteration as the reference. Consequently, the training objective for iterative DPO of the SLM at $m$-th iteration is formulated as:

$$\mathcal{L}_{\text{IDPO}}(\theta^{(m+1)}) := -\mathbb{E}_{(x,Q_+,Q_-)\sim\mathcal{D}_{\text{iDPO}}^{(m)}} \left[ \log \sigma \left( \beta \log \frac{\pi_\theta^{(m+1)}(Q_+|x)}{\pi_\theta^{(m)}(Q_+|x)} - \beta \log \frac{\pi_\theta^{(m+1)}(Q_-|x)}{\pi_\theta^{(m)}(Q_-|x)} \right) \right]. \tag{6}$$

# 5 Experiments

## 5.1 Experiment Setups

**Evaluation Datasets.** We use five representative multi-hop QA datasets: (1) **HotpotQA** (Yang et al., 2018), (2) **2WikiMQA** (Ho et al., 2020), (3) **MusiQue** (Trivedi et al., 2022), (4) **StrategyQA** (Geva et al., 2021), and (5) **Bamboogle** (Press et al., 2023). We conduct evaluations on all questions from StrategyQA and Bamboogle, and the first 500 questions from the development sets of the other datasets following existing studies (Trivedi et al., 2023; Shao et al., 2023; Wang et al., 2024). For Bamboogle, we used the Wikipedia dump from December 2018 as the corpus, while for the other datasets, we use the corpora provided by their respective original sources. Detailed descriptions are in Appendix A.

**Training Datasets.** We sample 10000 (question, answer) pairs from the training set of HotpotQA, MusiQue, and 2WikiMQA as the training data[4].

**Evaluation Metrics.** Different studies often use various metrics to evaluate the performance of RAG models. To ensure a comprehensive evaluation, we consider *Exact Match (EM)*, *Accuracy* and *F1 Score* jointly as the metric, while EM is used as the main metric.

**Baselines.** We consider the following baselines for comparison: (1) **White-box LLMs with RAG**: where we consider most recent RAG models based on open-source LLMs including DRAGIN (Su et al., 2024), GenGround (Shi et al., 2024b), ChatQA (Liu et al., 2024b), RankRAG (Yu et al., 2024b), and Retrieval-augmented Finetuning (RAFT) (Zhang et al., 2024b; Lin et al., 2024). (2) **Black-box LLMs with RAG**: where we consider IRCOT (Trivedi et al., 2023), FLARE (Jiang et al., 2023), RA-ISF (Liu et al., 2024a), BlendFilter (Wang et al., 2024), Search-o1 (Li et al., 2025), and IterDRAG (Yue et al., 2025). (3) **Additional Baselines:** we also consider baselines including Chain-of-Thought (COT, Wei et al. (2022)), vanilla RAG (Lewis et al., 2020), RAG with question decomposition (Khot et al., 2023), RAG with reranking, RAFE (Mao et al., 2024), Iter-RetGen (Shao et al., 2023), RQ-RAG (Chan et al., 2024), RAG-Star (Jiang et al., 2024), and one recent baseline RAG-Gym (Xiong et al., 2025). The details of baselines are in Appendix B.

## 5.2 Implementation Details

**Backbones.** For both white-box SLMs and black-box LLMs, we consider different variants to test the generalization of `Collab-RAG`. Specifically, we consider `Qwen-2.5-3B-Instruct` (Yang

---

[4]It is worth noting that we *do not* use the intermediate reasoning chain, or gold passages provided by the original datasets during our training and evaluation to ensure fair comparison.

| Baselines | StrategyQA | HotpotQA | | | MusiQue | | | 2WikiMQA | | | Bamboogle | | |
|---|---|---|---|---|---|---|---|---|---|---|---|---|---|
| | EM | Acc | EM | F1 | Acc | EM | F1 | Acc | EM | F1 | Acc | EM | F1 |
| **RAG with White-box LLMs (For Reference Only)** | | | | | | | | | | | | | |
| DRAGIN (Su et al., Best) | 68.9 | — | 31.4 | 42.3 | — | — | — | — | 30.4 | 39.3 | — | — | — |
| GenGround (Shi et al., 7B) | 77.1 | 47.2 | — | 52.2 | 20.2 | — | 27.4 | 45.6 | — | 50.2 | — | — | — |
| ChatQA (Liu et al., 70B) | — | — | 42.2 | 54.4 | — | — | — | — | 34.9 | 37.4 | — | — | — |
| RankRAG (Yu et al., 70B) | — | — | 42.7 | 55.4 | — | — | — | — | 38.2 | 43.9 | — | — | — |
| RAFT (Qwen-2.5-Instruct, 3B)* | 61.1 | 47.0 | 38.2 | 48.1 | 15.8 | 11.0 | 20.7 | 27.4 | 36.4 | 42.1 | 23.2 | 16.8 | 25.5 |
| RAFT (Llama-3.1-Instruct, 8B)* | 69.0 | 51.2 | 41.0 | 51.6 | 22.0 | 13.8 | 24.0 | 44.6 | 39.4 | 45.8 | 30.4 | 24.8 | 34.1 |
| **RAG with Black-box LLMs (For Reference Only)** | | | | | | | | | | | | | |
| FLARE (Jiang et al., GPT-3.5) | 77.3 | — | — | — | — | — | — | — | 51.0 | 59.7 | — | — | — |
| RA-ISF (Liu et al., Best) | 75.9 | — | 46.5 | — | — | — | — | — | 36.1 | — | — | — | — |
| BlendFilter (Wang et al., GPT-3.5) | 74.4 | — | 50.8 | 62.4 | — | — | — | — | 40.4 | 47.0 | — | — | — |
| Search-o1 (Li et al., 32B) | — | — | 45.2 | 57.3 | — | 16.6 | 28.2 | — | 58.0 | 71.4 | — | 56.0 | 67.8 |
| IterDRAG (Yue et al., Gemini-1.5, 32K) | — | 44.4 | 38.4 | 49.8 | 30.6 | 22.6 | 35.0 | 76.8 | 67.0 | 75.2 | 56.8 | 44.3 | 54.6 |
| **GPT-4o-mini as LLM Reader** | | | | | | | | | | | | | |
| CoT (Wei et al.) | 78.2 | 37.2 | 25.6 | 37.7 | 18.4 | 9.6 | 21.6 | 33.8 | 27.6 | 34.5 | 39.7 | 28.8 | 42.4 |
| RAG (Lewis et al.) | 78.6 | 58.4 | 41.8 | 57.1 | 22.6 | 11.4 | 23.4 | 45.0 | 37.2 | 45.9 | 28.0 | 23.2 | 32.5 |
| RAG w/ GPT-4o-mini Decompose (Khot et al.) | 76.8 | 61.6 | 46.6 | 60.4 | 41.4 | 24.2 | 39.3 | 74.0 | 59.0 | 71.5 | 57.6 | 45.6 | 60.6 |
| RAG w/ GPT-4o-mini Rerank | 76.8 | 56.0 | 40.2 | 56.5 | 23.2 | 12.4 | 25.3 | 44.2 | 36.8 | 44.7 | 34.4 | 28.0 | 37.0 |
| RAFE (Mao et al.) | 79.0 | 57.8 | 40.6 | 55.4 | 20.2 | 11.8 | 23.8 | 43.4 | 36.2 | 39.3 | 30.4 | 24.0 | 31.5 |
| IRCoT-0 shot (Trivedi et al.) | 77.7 | 58.8 | 44.8 | 57.6 | 31.4 | 18.8 | 31.2 | 54.8 | 43.6 | 56.6 | 37.6 | 28.8 | 41.6 |
| IRCoT-10 shot (Trivedi et al.) | 79.0 | 62.4 | 49.6 | 61.8 | 42.4 | 25.0 | 40.5 | 73.6 | 59.4 | 70.3 | 53.2 | 49.6 | 61.8 |
| Iter-RetGen (Shao et al.) | 73.0 | 60.0 | 46.2 | 61.2 | 39.6 | 25.8 | 40.4 | 65.8 | 50.2 | 65.3 | 50.4 | 40.8 | 51.0 |
| RQ-RAG (Chan et al.)†‡ | 73.5 | 60.2 | 45.4 | 59.5 | 31.0 | 19.2 | 31.0 | 59.6 | 50.2 | 58.8 | 25.2 | 14.8 | 21.5 |
| RAG-Star (Jiang et al.) | 69.0 | 49.0 | 46.0 | 60.0 | 27.0 | 22.0 | 30.7 | 43.0 | 38.0 | 46.8 | — | — | — |
| RAG-Gym (Xiong et al.)† | — | — | 46.4 | 59.4 | — | — | — | — | 49.4 | 58.0 | — | 55.2 | 65.6 |
| Collab-RAG 3B (Qwen-2.5-3B as backbone) | 82.0 | 67.2 | 51.6 | 66.2 | 41.6 | 25.4 | 39.6 | 79.4 | 63.0 | 74.5 | 55.4 | 47.2 | 62.0 |
| Collab-RAG 8B (Llama-3.1-8B as backbone) | 81.6 | 66.2 | 53.0 | 65.6 | 45.8 | 26.4 | 42.4 | 79.0 | 63.2 | 74.6 | 59.4 | 52.8 | 64.8 |
| **GPT-4o as LLM Reader** | | | | | | | | | | | | | |
| CoT (Wei et al.) | 79.5 | 56.2 | 29.4 | 48.9 | 26.8 | 17.0 | 28.9 | 58.4 | 41.8 | 53.6 | 60.8 | 42.8 | 59.5 |
| RAG (Lewis et al.) | 81.2 | 64.0 | 47.2 | 63.6 | 29.8 | 17.4 | 30.1 | 57.8 | 45.8 | 57.1 | 35.2 | 27.2 | 37.2 |
| RAG w/ GPT-4o Decompose (Khot et al.) | 83.2 | 67.2 | 52.2 | 65.6 | 44.8 | 27.8 | 42.3 | 78.8 | 62.2 | 73.3 | 69.6 | 62.4 | 74.9 |
| RAG w/ GPT-4o Rerank | 80.8 | 60.6 | 46.4 | 59.5 | 27.4 | 15.6 | 27.1 | 49.4 | 42.2 | 50.2 | 43.2 | 29.6 | 42.6 |
| RAFE (Mao et al.) | 78.6 | 61.0 | 44.8 | 58.8 | 22.8 | 13.8 | 23.2 | 50.6 | 43.2 | 44.1 | 35.2 | 26.4 | 36.9 |
| IRCoT-0 shot (Trivedi et al.) | 78.6 | 64.2 | 48.0 | 63.7 | 33.8 | 22.4 | 33.5 | 61.4 | 51.4 | 61.0 | 60.8 | 46.4 | 56.9 |
| IRCoT-10 shot (Trivedi et al.) | 81.2 | 66.4 | 52.8 | 66.0 | 44.2 | 29.4 | 43.9 | 78.0 | 62.2 | 70.0 | 66.4 | 57.6 | 70.0 |
| Iter-RetGen (Shao et al.)† | 76.8 | 62.6 | 48.4 | 63.4 | 42.0 | 26.6 | 42.6 | 71.4 | 52.8 | 69.6 | 62.4 | 48.8 | 67.7 |
| RQ-RAG (Chan et al.)†‡ | 77.3 | 62.0 | 46.2 | 60.3 | 31.6 | 20.0 | 32.4 | 60.2 | 50.8 | 59.5 | 32.8 | 23.6 | 33.9 |
| RAG-Star (Jiang et al.)† | 81.0 | 57.0 | 48.0 | 68.6 | 40.0 | 29.0 | 43.5 | 63.0 | 48.0 | 61.7 | — | — | — |
| Collab-RAG 3B (Qwen-2.5-3B as backbone) | 82.5 | 68.6 | 55.6 | 68.3 | 43.6 | 26.2 | 40.0 | 82.0 | 67.0 | 77.9 | 65.6 | 60.0 | 70.6 |
| Collab-RAG 8B (Llama-3.1-8B as backbone) | 82.9 | 69.2 | 54.4 | 68.3 | 47.2 | 29.0 | 43.4 | 81.0 | 67.2 | 77.0 | 69.6 | 63.2 | 74.0 |

Table 1: Comparison of various baselines on multiple datasets. Baselines with specific sizes have been annotated in parentheses. *: Baselines require gold passage annotation labels. †: Baselines require distillation from frontier LLMs (e.g. GPT-4 or GPT-4o). ‡: The original paper studied over *reading comprehension setting* where gold passages are given, which is different from the setting of this study.

et al., 2024) and Llama-3.1-8B-Instruct (Dubey et al., 2024) as white-box SLMs, and consider GPT-4o-mini and GPT-4o (Hurst et al., 2024) as black-box LLMs during evaluation. In the model training stage, we only use GPT-4o-mini as the LLM reader.

**Hyperparameters.** Training Collab-RAG is conducted on eight NVIDIA A100 GPUs. We employ the AdamW optimizer with a learning rate of 2e-6 for both the Qwen-2.5-3B and Llama-3.1-8B models during the warmup stage, and 1e-6/5e-7 for Qwen-2.5-3B and Llama-3.1-8B, respectively, in the preference optimization stage. The batch size is set to 64. We set $\beta = 0.5, N = 5$ in iterative preference optimization by default. For retrieval setup, we use the Dragon-Plus[5] (Lin et al., 2023) as the retriever and set the number of retrieved passage $k$ to 10[6]. To ensure a fair comparison, we use the same retriever for baselines and test the performance of $k \in \{5, 10, 15, 20\}$ and report *the best performance* for baselines. For generation, we use greedy sampling and set the max number of generated tokens to 64.

## 5.3 Main Experiment Results

Table 1 compares Collab-RAG and baselines. We have the following observations:

- Collab-RAG exhibits strong empirical performance: With a lightwighted GPT-4o-mini as the backbone, it outperforms existing black-box and white-box retrieval-augmented language models on complex QA tasks by 14.2% and 6.6% on average.

---

[5]https://huggingface.co/facebook/dragon-plus-context-encoder
[6]We study the effect of different $k$ and $\beta$ in Appendix C.1.

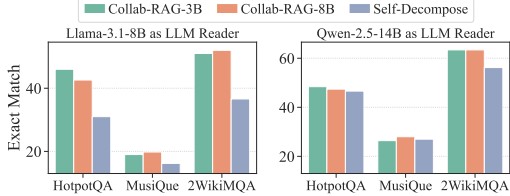
Figure 3: Different LLM Readers

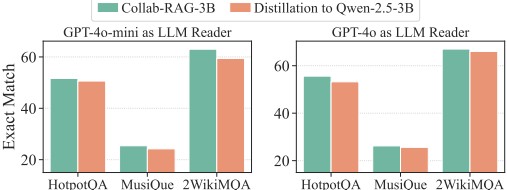
Figure 4: `Collab-RAG` v.s. Distillation

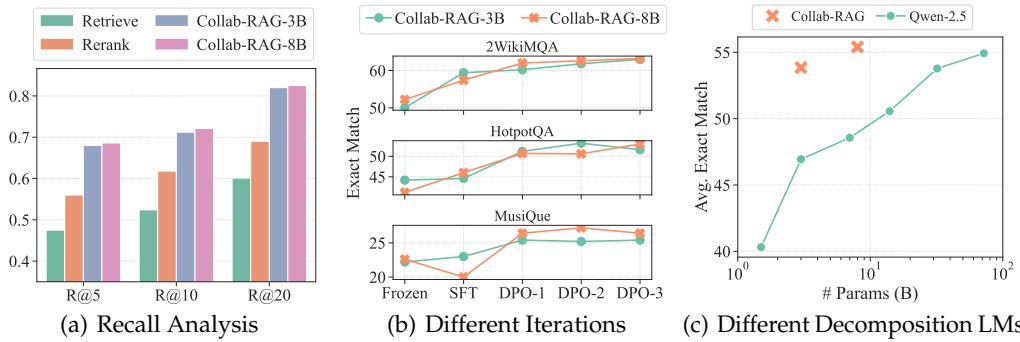
(a) Recall Analysis      (b) Different Iterations      (c) Different Decomposition LMs

Figure 5: Additional Studies. GPT-4o-mini as the default LLM reader.

- `Collab-RAG` is efficient: Using an 8B model for question decomposition, it outperforms LLM-based decomposition baselines on most datasets (5/5 for GPT-4o-mini, 4/5 for GPT-4o) in EM, without requiring GPT-4o distillation. With a 3B model, it surpasses the GPT-4o-based decomposer with an average gain of 0.7%.

- `Collab-RAG` achieves competitive performance against recent baselines: It surpasses RAG-Gym on 2/3 datasets and RAG-Star on 3/4 datasets. Note that these baselines are based on process reward models and inference-time search algorithms. Since our contributions are orthogonal to these methods, they have the potential to be combined for further improvements.

### 5.4 Ablation Studies

**Effect of Different Designs in Finetuning SLMs.** We analyze the impact of different components in Table 2 with GPT-4o-mini as LLM reader. Notably, incorporating a relaxed reward (e.g., accuracy-based reward) and a format-based reward contributes to improved EM performance across target datasets. Additionally, relying solely on single-step SFT and DPO results in significant performance degradation, particularly when using Llama-3-8B as the backbone, highlighting the necessity of iterative preference optimization.

**Different LLM and Retriever Backbones.** Figure 3 presents the performance of `Collab-RAG` with various LLM readers (Llama-3.1-8B-It (Dubey et al., 2024) and Qwen-2.5-14B-It (Yang et al., 2024)) and Table 3 shows the performance with different retrievers (COCO-DR (Yu et al., 2022), GTE (Li et al., 2023) and E5 (Wang et al., 2022)) when GPT-4o-mini is used as the LLM reader. *For LLM backbones*, we observe that `Collab-RAG` provides significant gains, particularly when using a less powerful LLM reader (e.g., 10.7% for Llama-3.1-8B). *For retrievers*, `Collab-RAG` mostly outperforms baselines across different retrieval choices, demonstrating its robustness to varying retriever configurations.

**`Collab-RAG` v.s. Direct Distillation from Black-box LLMs.** We further study the performance of `Collab-RAG` with baselines fine-tuned on synthetic question decompositions generated by GPT-4o-mini and GPT-4o models after rejection sampling. As shown in Figure 4, `Collab-RAG` consistently outperforms SLM models trained through distillation. This suggests that simple distillation may not be the optimal solution for query decomposition.

| | HotpotQA EM | MusiQue EM | 2WikiMQA EM |
|---|---|---|---|
| Collab-RAG (Qwen-2.5-3B) | 51.6 | **25.4** | **63.0** |
| w/o Format Reward | 50.2 | 23.2 | 62.2 |
| w/o Accuracy Reward | 51.4 | 24.0 | **63.0** |
| w/o iterative DPO | **52.0** | 24.2 | 61.8 |
| SFT Only | 49.4 | 23.6 | 62.0 |
| Collab-RAG (Llama-3.1-8B) | **53.0** | **26.4** | 63.2 |
| w/o Format Reward | 49.4 | 24.2 | 62.4 |
| w/o Accuracy Reward | 52.0 | 24.8 | 62.4 |
| w/o iterative DPO | 47.6 | 24.4 | **64.2** |
| SFT Only | 47.0 | 22.6 | 61.8 |

Table 2: Different Designs in SLMs.

| | HotpotQA EM | MusiQue EM | 2WikiMQA EM |
|---|---|---|---|
| Collab-RAG (Qwen-2.5-3B) | 51.6 | 25.4 | 63.0 |
| w/ COCO-DR | 50.4 | 28.0 | 62.8 |
| w/ E5-Large | 52.2 | 25.2 | 64.6 |
| w/ GTE | 50.4 | 25.6 | 64.2 |
| Collab-RAG (Llama-3.1-8B) | 53.0 | 26.4 | 63.2 |
| w/ COCO-DR | 50.6 | 28.4 | 65.0 |
| w/ E5-Large | 52.0 | 28.4 | 65.2 |
| w/ GTE | 50.4 | 26.6 | 64.2 |
| Best Baseline in Table 1 | 46.6 | 25.8 | 59.0 |

Table 3: Different Retriever Backbones.

| Question | *Piers Bizony has written articles for which magazine based in Bristol, UK?* |
|---|---|
| **No Decomposition** | **Answer:** None |
| **GPT-4o-mini Self-decompose** | **Decomposed Question:** What magazine is based in Bristol, UK? **Answer:** I cannot identify a specific magazine based in Bristol from the information given. **Decomposed Question:** For which magazine has Piers Bizony written articles? **Answer:** The context provided does not mention any magazine based in Bristol, UK. |
| **Collab-RAG-3B** | **Decomposed Question:** What magazines has Piers Bizony written articles for? **Answer:** The Independent, BBC Focus, Wired **Decomposed Question:** Among the magazines mentioned in #1, which one is based in Bristol, UK? **Answer:** BBC Focus |

Table 5: A case study comparing GPT-4o-mini's direct answer w/o decomposition, its self-decomposed questions and answers, and the decomposed responses from Collab-RAG-3B.

## 5.5 Additional Studies

**Recall Analysis.** Figure 5(a) shows the passage-level answer recall on the HotpotQA dataset. We observe that while reranking can improve over retrieval, its gain is limited. Instead, for complex queries, decomposition serves as a more directly way to improve search quality.

**Performance with Different Iterations.** Figure 5(b) illustrates the training dynamics of Collab-RAG across different stages. The results show consistent performance gains during the first 1–2 DPO rounds. However, by stage 3, the improvement begins to plateau. Using 3 rounds strikes a balance between performance and efficiency.

**Performance with Different Question Decomposition LM.** We compare Collab-RAG against frozen Qwen-2.5-Instruct series LMs ranging from 1.5B to 72B on the question decomposition task, using GPT-4o-mini as the default LLM reader. As shown in Figure 5(c), Collab-RAG with a 3B backbone outperforms a frozen 32B LM (10.7× larger), and with a 8B backbone, it surpasses a frozen 72B LM (9× larger), highlighting its parameter efficiency.

**Performance with Different Preference Optimization Algorithms.** Apart from DPO, we also tried to use SimPO (Meng et al., 2024) and ORPO (Hong et al., 2024) as preference optimization algorithms, yet we observe that DPO yields most robust performance.

| Algorithm | HotpotQA | MusiQue | 2WikiMQA |
|---|---|---|---|
| Collab-RAG (Qwen-2.5-3B) | **51.6** | **25.4** | **63.0** |
| w/ ORPO | 48.8 | 24.4 | 62.2 |
| w/ SimPO | 47.6 | 24.2 | 62.8 |

Table 4: Performance of Different Optimization Algorithm with EM as the metric.

## 5.6 Case studies

Table 5 presents a case study comparing GPT-4o-mini's direct answer w/o decomposition, its self-decomposed questions and answers, and the decomposed responses from Collab-RAG-3B. We observe that, given a question from HotpotQA, GPT-4o-mini cannot infer an answer from the context retrieved directly from the original question. Even after self-decomposition, it still fails to find an answer as the first question is too broad, making it difficult for the retriever to retrieve relevant context. In contrast, Collab-RAG-3B decomposes the question in a more structured and human-like manner. This approach allows it to retrieve the right context step by step, ultimately leading to the correct answer.

# 6 Conclusion

We introduce Collab-RAG, a framework that fosters collaboration between a white-box SLM and a black-box LLM to enhance RAG for multi-hop question-answering. Through iterative DPO guided by supervision signals from an affordable black-box LLM (GPT-4o-mini), Collab-RAG significantly enhances the SLM's question decomposition capabilities without expensive human annotations or resource-intensive model distillation. Experimental results demonstrate that our training strategy consistently outperforms standard RAG models (14.2%) and strong decomposition-based baselines (1.8%) over 5 multi-hop QA datasets, exhibiting robust generalization across various black-box LLMs. Collab-RAG presents a scalable and efficient solution to improve complex retrieval-augmented question-answering scenarios. An important line of future work is to extend Collab-RAG for online reinforcement learning (Jin et al., 2025).

## Acknowledgments

This research was partially supported by the Texas Advanced Computing Center (TACC) and the NVIDIA Academic Grant Program to Dr. Wenqi Shi. RX and CY were partially supported by the US National Science Foundation under Award Numbers 2319449, 2312502, and 2442172, as well as the US National Institute of Diabetes and Digestive and Kidney Diseases of the US National Institutes of Health under Award Number K25DK135913. JH were partially supported by the US National Science Foundation (NSF) grant IIS-2145411.

## Reproducibility Statement

The five datasets used in this study are all publicly available. Detailed descriptions of these datasets and their corresponding tasks are provided in Appendix A. In section 5.1, we describe the experimental setup and key parameter settings. The implementation of Collab-RAG will be open-sourced upon acceptance.

## Ethics Statement

Collab-RAG involves the usage of OpenAI APIs. We follow the data usage guidelines for interactions with Microsoft Azure's OpenAI API service and opt out of the human review process by completing and submitting the Azure OpenAI Additional Use Case Form. We do not foresee other ethics issues.

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

## A  Dataset Details

Here are the details for each dataset used in our experiments:

- HotpotQA (Yang et al., 2018): A large-scale multi-hop QA dataset that requires reasoning over multiple Wikipedia passages to answer fact-based questions.
- 2WikiMQA (Ho et al., 2020): A dataset extending HotpotQA, featuring questions that require reasoning across two different Wikipedia articles, emphasizing cross-document retrieval.
- MuSiQue (Trivedi et al., 2022): A challenging multi-hop QA dataset with questions that explicitly require reasoning across multiple sentences scattered across different documents.
- StrategyQA (Geva et al., 2021): A dataset focusing on implicit reasoning, where answering requires strategic multi-step inference rather than direct fact lookup.
- Bamboogle (Press et al., 2023): A dataset designed to evaluate LLMs' ability to answer adversarial and compositional questions, requiring careful decomposition and reasoning.

## B  Baseline Details

Here are the details for each baseline used in our experiments:

- DRAGIN (Su et al., 2024): It dynamically determines retrieval timing and content by modeling the evolving information of the language model generation.
- GenGround (Shi et al., 2024b): It iteratively generates simpler single-hop questions and directly grounds their answers through retrieved external documents, combining internal model knowledge and external context to solve multi-hop questions.
- ChatQA (Liu et al., 2024b): It enhances retrieval-augmented conversational question answering via a two-stage instruction-tuning: initial fine-tuning on general instruction-following data followed by specialized context-enhanced tuning.
- RankRAG (Yu et al., 2024b): It unifies context ranking and answer generation into a single instruction-tuned LLM, by introducing a ranking-as-generation task during training, training the model to rerank retrieved contexts before generating answers.
- RAFT (Zhang et al., 2024b; Lin et al., 2024): It employs instruction tuning to train the LLM for producing chain-of-thought answers explicitly grounded in relevant retrieved contexts.
- IRCOT (Trivedi et al., 2023): It interleaves retrieval and chain-of-thought reasoning, progressively refining reasoning steps and associated retrieval queries based on previous outputs.
- FLARE (Jiang et al., 2023): It iteratively predicts upcoming sentences to actively decide when and what information to retrieve during generation, enabling continuous retrieval based on anticipated information needs.
- RA-ISF (Liu et al., 2024a): It iteratively decomposes complex tasks and integrates self-feedback mechanisms across submodules, refining retrieval and generation steps to minimize irrelevant contexts.
- BlendFilter (Wang et al., 2024): It iteratively blends query generation with knowledge filtering, employing LLM-generated feedback to dynamically eliminate irrelevant information from retrieved contexts.
- Search-o1 (Li et al., 2025): It uses an agentic retrieval mechanism that dynamically retrieves external information upon encountering uncertain reasoning steps and employs a dedicated Reason-in-Documents module to filter out irrelevant details.
- IterDRAG (Yue et al., 2025): It scales inference through iterative retrieval and generation steps, employing flexible test-time strategies such as increasing retrieved

documents or generation steps, thereby improving the effective utilization of contextual information in long-context reasoning tasks.

- COT (Wei et al., 2022): It enhances LLM's reasoning abilities by guiding them to generate intermediate reasoning steps before generating the final answer.

- RAG (Lewis et al., 2020): It retrieves top passages from the corpus before generating the answer.

- RAG with question decomposition (Khot et al., 2023): It uses the same LLM as the reader model to break down complex questions into simpler sub-questions, then retrieves relevant information for each, and synthesizes the answers to address the original question.

- RAG with reranking: It uses the same LLM as the reader model to rerank the top-retrieved passages before generating the final answer.

- RAFE (Mao et al., 2024): It rewrites questions based on the LLM ranking feedback.

- Iter-RetGen (Shao et al., 2023): It interleaves retrieval and generation processes in multiple iterations, allowing the model to refine its queries and responses progressively for better accuracy.

- RQ-RAG (Chan et al., 2024): It explicitly enhances RAG by training the model to refine, rewrite, decompose, and disambiguate complex queries, addressing limitations in ambiguous or insufficiently detailed original queries.

- RAG-Star (Jiang et al., 2024): It integrates RAG with Monte Carlo Tree Search, iteratively using retrieved information to guide and improve the tree-based deliberative reasoning process of language models.

- RAG-Gym (Xiong et al., 2025): It employs fine-grained, step-wise process supervision and trained reward models to iteratively optimize the retrieval and generation processes.

## C  Additional Experimental Results

### C.1  Parameter Study

Table 6 shows the result of `Collab-RAG` with different $\beta$ and $k$. From the result, we observe that `Collab-RAG` is not sensitive to the selection of $\beta$. Besides, setting $K$ too large or too small will lead to performance degradation. This is because too small $k$ yields limited recall, yet too large $k$ can introduce many irrelevant information.

Table 6: Performance of `Collab-RAG` with Qwen-2.5-3B on different benchmarks under varying $\beta$ and $k$ settings.

| Parameter | HotpotQA | MusiQue | 2WikiMQA |
|---|---|---|---|
| $\beta = 0.5, k = 5$ | 50.8 | 24.6 | 60.6 |
| $\beta = 0.5, k = 10$ (Our Setting) | 51.6 | 25.4 | 63.0 |
| $\beta = 0.5, k = 15$ | 51.4 | 25.8 | 62.4 |
| $\beta = 0.1, k = 5$ | 50.6 | 24.8 | 60.2 |
| $\beta = 0.1, k = 10$ | 51.0 | 25.2 | 61.4 |
| $\beta = 0.1, k = 15$ | 50.4 | 24.6 | 59.8 |

### C.2  Performance with Different Size of White-box Query Decomposer

Table 7 lists detailed performance on each dataset for directly prompting frozen white-box LMs for query decomposition, using GPT-4o-mini as the LLM reader.

Table 7: Performance of LLM query decomposers across multiple QA datasets.

| Model | StrategyQA Acc | HotpotQA Acc | EM | F1 | MusiQue Acc | EM | F1 | 2WikiMQA Acc | EM | F1 | Bamboogle Acc | EM | F1 |
|---|---|---|---|---|---|---|---|---|---|---|---|---|---|
| Qwen-2.5-1.5B-instruct | 74.2 | 44.2 | 32.4 | 43.2 | 25.4 | 16.4 | 25.9 | 49.4 | 40.2 | 49.2 | 46.4 | 38.4 | 51.7 |
| Qwen-2.5-3B-instruct | 75.1 | 55.8 | 44.2 | 56.1 | 40.8 | 22.2 | 37.5 | 62.0 | 50.0 | 60.0 | 52.0 | 43.2 | 53.8 |
| Qwen-2.5-7B-instruct | 76.0 | 61.2 | 47.2 | 60.3 | 35.8 | 22.6 | 36.0 | 65.2 | 53.8 | 64.6 | 53.6 | 43.2 | 54.1 |
| Qwen-2.5-14B-instruct | 74.2 | 64.6 | 50.4 | 63.6 | 40.6 | 25.8 | 39.2 | 71.8 | 56.8 | 68.0 | 56.0 | 45.6 | 57.7 |
| Qwen-2.5-32B-instruct | 79.9 | 64.8 | 52.0 | 65.1 | 40.2 | 25.4 | 39.2 | 78.8 | 62.0 | 73.6 | 60.0 | 49.6 | 63.1 |
| Qwen-2.5-72B-instruct | 78.6 | 68.6 | 55.6 | 68.3 | 43.6 | 26.2 | 39.9 | 83.0 | 67.0 | 77.9 | 56.0 | 47.2 | 64.8 |

# D Prompt Details

The detailed prompts for query decomposition as well as question answering is listed in the following. It is worth noting that for the decomposition step, we select two demonstrations from the training set as the input to the LLM for *both* our method and baselines to enable a fair comparison.

Listing 1: Prompts for Query Decomposition

```
Please break down the given question into multiple specific sub-questions
 that address individual components of the original question.
Please generate the decomposed sub-questions for the below question. The
sub-question should be labeled with a reference to previous answers (e.g
., #1) when needed. For example, #1 means the answer for decomposed
question 1.

Here are two examples:
[[Begin of the Example 1]]
## Question:
What is the average winter daytime temperature in the region containing
Richmond, in the state where WXBX is located?

## Decomposed Question:
### Q1: Which state is WXBX located?
### Q2: In which of #1 's regions is Richmond?
### Q3: What is the average winter daytime temperature in #2?
[[End of the Example 1]]

[[Begin of the Example 2]]
## Question:
How long was the place where the Yongle Emperor greeted the person to
whom the edict was addressed the capitol of the area where Guangling
District was located?

## Decomposed Question:
### Q1: Who was the edict addressed to?
### Q2: Where did the Yongle Emperor greet #1 ?
### Q3: Where does Guangling District locate?
### Q4: How long had #2 been the capital city of #3 ?
[[End of the Example 2]]
Now, decompose the following question:
## Question:
[question]

## Decomposed Question:
```

Listing 2: Prompts for Answering Subquestions

```
You have the following context passages:
[Retrieve top-k Context]

Please answer the question '[subquestion]' with a short span using the
context as reference.
```

```
If no answer is found in the context, use your own knowledge.
Do not give any explanation. Your answer needs to be as short as possible
.
```

Listing 3: Prompts for generating the final answer for Question Answering

```
For the question: [original question]

We have the following decomposed sub-questions and sub-answers:
[subquestion and answers]

Based on these, provide the final concise answer to the original question
. Do not give an explanation.
```

