# OpenReview forum: "Collab-RAG: Boosting Retrieval-Augmented Generation for Complex Question Answering via White-Box and Black-Box LLM Collaboration"
_colmweb.org/COLM/2025/Conference — COLM 2025_

### Official Review · Reviewer_k4rg · 2025-05-12

**Rating:** 6
**Confidence:** 3
**Ethics Flag:** 1

**Summary:**

This paper presents an approach to complex QA, Collab-RAG, that relies on a black-box LLM for providing training signal to a smaller white-box LLM. This is done by performing iterative preference optimization after a supervised fine-tuning warm up step. The experimental results show that the proposed approach is promising, however I have concerns regarding the experimental setup and presentation issues.

**Reasons To Accept:**

* Interesting approach to complex QA that utilizes iterative preference optimization as opposed to knowledge distillation.
* Outperforms baselines under comparison.

**Reasons To Reject:**

* The paper presentation could be improved. In Section 4 the training and inference stages are not clearly separated; having separate (sub)sections would help the reader. Also, the framing of the RAG system as an environment in Section 4.2. does not necessarily help in formulating the approach.
* The compared baselines use different and sometimes older backbone architectures than the ones used in Collab-RAG; also, it is not clear which training data the baselines used and how that relates to the ones Collab-RAG uses. These raise questions regarding the fairness of the comparisons.
* The claim that this approach is better than knowledge distillation is not convincing; the only evidence provided is in Figure 4, where the differences seem small, and important experimental details are missing, such as how much data have been generated by the teacher model, how does that training set compare to the one used by Collab-RAG, as well as an analysis on the added complexity of having to implement this framework compared to the (simpler) distillation framework.
* The paper would benefit from quantitative/qualitative analysis on the errors of the model, and also discussion on why the proposed method is under-performed by baselines in Bamboogle.
* Lacking discussion on relevant related work / baselines [1,2,3]

[1] Jeong et al. Adaptive-RAG: Learning to Adapt Retrieval-Augmented Large Language Models through Question Complexity. ACL 2024
[2] Huang et al. Question Decomposition Tree for Answering Complex Questions over Knowledge
Bases. AAAI 2023
[3] Fu et al. Decomposing Complex Questions Makes Multi-Hop QA Easier and More Interpretable. EMNLP 2021

---

> ### Author Response · Authors · 2025-06-02
> **Initial Rebuttal to Reviewer k4rg (Part 3)**
>
> > W5: Lacking discussion on relevant related work / baselines [1,2,3]
> [1] Jeong et al. Adaptive-RAG: Learning to Adapt Retrieval-Augmented Large Language Models through Question Complexity. ACL 2024 [2] Huang et al. Question Decomposition Tree for Answering Complex Questions over Knowledge Bases. AAAI 2023 [3] Fu et al. Decomposing Complex Questions Makes Multi-Hop QA Easier and More Interpretable. EMNLP 2021
>
> A: Thanks for mentioning these related papers, we will discuss them in the next version of the paper. Here is an empirical comparison of Collab-RAG and Adaptive RAG using GPT-4o-mini as the reader:
>
> | Method                         | StrategyQA |        |          | |     HotpotQA   |        | |    MusiQue    |        | |     2WikiMQA   |        | |       Bamboogle |        |
> |--------------------------------|------------|--------|----------|----------|--------|--------|---------|--------|--------|-----------|--------|--------|------------|--------|--------|
> |                                | Acc        |        |          | Acc      | EM     | F1     | Acc     | EM     | F1     | Acc       | EM     | F1     | Acc        | EM     | F1     |
> | AdaptiveRAG (Best reported in the paper)             | ---        |        |          | 46.8     | 44.2   | 54.8   | 29.6    | 21.8   | 32.6   | 54.0      | 47.6   | 57.4   | --         | --     | --     |
> | AdaptiveRAG (ours, GPT-4o-mini Reader)      | 78.6       |        |          | 52.4     | 46.4   | 58.1   | 30.2    | 21.4   | 31.8   | 59.2      | 51.4   | 60.9   | 58.4       | 44.8   | 55.1   |
> | Collab-RAG w/ Qwen-2.5-3B      | 82.0       |        |          | 67.2     | 51.6   | 66.2   | 41.6    | 25.4   | 39.6   | 79.4      | 63.0   | 74.5   | 55.4       | 47.2   | 62.0   |
> | Collab-RAG w/ Llama-3.1-8B     | 81.6       |        |          | 66.2     | 53.0   | 65.6   | 45.8    | 26.4   | 42.4   | 79.0      | 63.2   | 74.6   | 59.4       | 52.8   | 64.8   |
>
> In addition, we note that papers [2,3] are loosely related, as they also investigate question decomposition. However, paper [2] focuses on *question answering over structured knowledge bases*, which differs significantly from our open-domain QA setting. Paper [3] relies on *annotated metadata such as question types, relations, and subjects*, whereas Collab-RAG operates with supervision only from final answers, requiring no intermediate annotations. Therefore, we believe these works are not directly comparable to our approach, but we will make sure to mention them in the related works.
>
> ***
> Thank you once again for your helpful review. We hope our response could address your concerns. If you have any further questions, we would be happy to discuss them further.

---

> > ### Comment · Reviewer_k4rg · 2025-06-08
> >
> > Thank you for the response, I have increased my score.

---

> > > ### Author Response · Authors · 2025-06-11
> > >
> > > Thank you for taking time to re-evaluate our paper. We appreciate it!

---

> ### Author Response · Authors · 2025-06-02
> **Initial Rebuttal to Reviewer k4rg (Part 2)**
>
> > W3: The claim that this approach is better than knowledge distillation is not convincing; the only evidence provided is in Figure 4, where the differences seem small, and important experimental details are missing, such as how much data have been generated by the teacher model, how does that training set compare to the one used by Collab-RAG, as well as an analysis on the added complexity of having to implement this framework compared to the (simpler) distillation framework.
>
> A: Thank you for the thoughtful feedback. We clarify that our comparison to knowledge distillation is conducted under controlled and fair conditions – **using the same prompts and a comparable number of training samples**. Specifically, the distillation setup involves approximately 15k examples generated by a teacher model (e.g., GPT-4o-mini), whereas Collab-RAG is trained with around 8k samples for SFT and ~2k for each stage of DPO, totaling a similar data scale.
>
> It's also important to note that for the distillation process, we apply techniques similar to us such as rejection sampling to filter out low-quality decompositions (see L299). In fact, naively using the raw decompositions from GPT-4o-mini without filtering yields poor performance, e.g., only ~18% EM on MusiQue, ~54% on 2WikiMQA and ~42% EM on HotpotQA, significantly lower than the distillation results reported in Figure 4.
>
> These observations suggest that our joint training approach is necessary for improving the end task performance: **it offers stronger performance and robustness than distillation, even under similar data and model budgets**. We will update the manuscript to clarify these points and include more implementation details for transparency.
>
> ***
> > W4: The paper would benefit from quantitative/qualitative analysis on the errors of the model, and also discussion on why the proposed method is under-performed by baselines in Bamboogle.
>
> A: Thank you for the suggestion. We conducted a manual error analysis and identified two primary sources of failure:
> - **Decomposition challenges on complex questions**: For examples requiring ≥4 reasoning hops, the model often struggles with accurate intermediate decomposition. Expanding training data to better cover such complex cases may improve performance.
>
>
> - **Limited context utilization by black-box LLMs**: Even when relevant information is retrieved, black-box LLMs sometimes fail to extract it precisely—especially from long or noisy contexts. This suggests that additional RAG-oriented fine-tuning on the reader could further enhance answer accuracy.
>
>
> We attribute the relatively lower performance on Bamboogle (when compare to baselines such as Search-o1 and RAG-Gym) to two main factors:
> - **Retrieval limitations**: Our experiments use the standard Wikipedia corpus for retrieval, which does not cover some of the queries in Bamboogle. In contrast, Search-o1 leverages commercial search engines that can retrieve more comprehensive and up-to-date information.
> - **Training supervision**: RAG-Gym surpasses our method when using GPT-4o-mini as the reader because it relies on process-level feedback generated by a stronger model (GPT-4o) for data creation. In contrast, our method is trained purely with GPT-4o-mini supervision. However, on HotpotQA and 2WikiMQA, our approach yields significantly higher gains (5.2%–13.6%), and overall achieves better average performance across these three benchmarks. Note that RAG-Gym is not evaluated on StrategyQA or MusiQue.
>
> We will incorporate these insights into the revised manuscript and expand our discussion on the performance gap in Bamboogle.

---

> ### Author Response · Authors · 2025-06-02
> **Initial Rebuttal to Reviewer k4rg (Part 1)**
>
> We sincerely thank the reviewer for the helpful feedback. Please find our responses to each of the comments below.
>
> ***
> > W1: The paper presentation could be improved.
>
> A: We thank the reviewer for these helpful suggestions. We will follow your suggestions to separate the training and inference stages and better formulate our method to improve the readability of the paper in the next version.
>
> ***
> > W2: The compared baselines use different and sometimes older backbone architectures than the ones used in Collab-RAG; also, it is not clear which training data the baselines used and how that relates to the ones Collab-RAG uses. These raise questions regarding the fairness of the comparisons.
>
> A: Thanks for pointing this out. We want to highlight that some of the baselines (e.g. BlendFilter, IterDRAG) are not open-sourced, and we mainly compare with them for reference. That being said, per your suggestion, we have tried our best to reproduce and report the results of representative baselines (IRCoT in zero-shot and 10-shot setting, BlendFilter, and IterDRAG) using the same backbones GPT-4o-mini and GPT-4o as Collab-RAG. The results are shown in the following table:
>
> | Baseline                     | StrategyQA |        |        |  |    HotpotQA    |       |  |     MusiQue   |       |   |    2WikiMQA   |       |  |  Bamboogle     |       | Avg. EM       |
> |-----------------------------|------------|--------|--------|----------|--------|-------|---------|--------|-------|-----------|-------|-------|------------|-------|-------|----------------|
> |                             | EM         |        |        | Acc      | EM     | F1    | Acc     | EM     | F1    | Acc       | EM    | F1    | Acc        | EM    | F1    |                |
> | **GPT-4o-mini as reader LLM** |            |        |        |          |        |       |         |        |       |           |       |       |            |       |       |                |
> |IRCoT-zero shot              | 77.7       |        |        | 58.8     | 44.8   | 57.6  | 31.4    | 18.8   | 31.2  | 54.8      | 43.6  | 56.6  | 37.6       | 28.8  | 41.6  | 42.7           |
> | IRCoT-10 shot              | 79.0       |        |        | 62.4     | 49.6   | 61.8  | 42.4    | 25.0   | 40.5  | 73.6      | 59.4  | 70.3  | 53.2       | 49.6  | 61.8  | 52.5           |
> | BlendFilter                 | 75.9       |        |        | 59.6     | 45.6   | 59.9  | 24.8    | 12.8   | 29.3  | 56.2      | 44.6  | 55.7  | 60.8       | 52.8  | 61.2  | 46.3           |
> | IterDRAG                    | 78.6       |        |        | 60.6     | 46.8   | 59.8  | 27.8    | 19.6   | 28.1  | 74.2      | 63.6  | 71.7  | 54.0       | 46.4  | 60.7  | 51.0           |
> | **Collab-RAG w/ Qwen-2.5-3B**  | 82.0       |        |        | 67.2     | 51.6   | 66.2  | 41.6    | 25.4   | 39.6  | 79.4      | 63.0  | 74.5  | 55.4       | 47.2  | 62.0  | **53.8 (+2.5%)** |
> | **Collab-RAG w/ Llama-3.1-8B** | 81.6       |        |        | 66.2     | 53.0   | 65.6  | 45.8    | 26.4   | 42.4  | 79.0      | 63.2  | 74.6  | 59.4       | 52.8  | 64.8  | **55.4 (+5.5%)** |
> | **GPT-4o as reader LLM**    |            |        |        |          |        |       |         |        |       |           |       |       |            |       |       |                |
> | IRCoT-zero shot              | 78.6       |        |        | 64.2     | 48.0   | 63.7  | 33.8    | 22.4   | 33.5  | 61.4      | 51.4  | 61.0  | 60.8       | 46.4  | 56.9  | 49.4           |
> | IRCoT-10 shot              | 81.2       |        |        | 66.4     | 52.8   | 66.0  | 44.2    | 29.4   | 43.9  | 78.0      | 62.2  | 72.9  | 66.4       | 57.6  | 70.0  | 56.6           |
> | BlendFilter                 | 78.6       |        |        | 64.4     | 51.6   | 64.7  | 36.2    | 19.6   | 36.5  | 77.2      | 59.6  | 72.4  | 67.2       | 59.2  | 70.4  | 53.7           |
> | IterDRAG                    | 81.2       |        |        | 64.2     | 51.2   | 65.1  | 38.8    | 25.0   | 37.9  | 78.8      | 68.4  | 77.2  | 62.0       | 55.2  | 67.8  | 56.2           |
> | **Collab-RAG w/ Qwen-2.5-3B**  | 82.5       |        |        | 68.6     | 55.6   | 68.3  | 43.6    | 26.2   | 40.0  | 82.0      | 67.0  | 77.9  | 66.4       | 60.0  | 70.6  | **58.3 (+3.0%)** |
> | **Collab-RAG w/ Llama-3.1-8B** | 82.9       |        |        | 69.2     | 54.4   | 68.3  | 47.2    | 29.0   | 43.4  | 81.0      | 67.2  | 77.0  | 69.6       | 63.2  | 74.0  | **59.3 (+4.8%)** |
>
> From these results, we observe that Collab-RAG consistently outperforms all baselines when using the same LLM reader (e.g., GPT-4o-mini or GPT-4o).
>
>
> Regarding training-based baselines such as Iter-RetGen and RQ-RAG, we ensure fair comparisons by **training them on the same datasets** used for Collab-RAG (i.e., HotpotQA, 2WikiMQA, and MusiQue). We will clarify this in the revised manuscript for full transparency.

---

### Official Review · Reviewer_YbMU · 2025-05-13

**Rating:** 7
**Confidence:** 4
**Ethics Flag:** 1

**Summary:**

This paper introduces Collab-RAG, a new framework that allows 2 LLMs to collaborate: one is large and black-box, and the other is small and open-box. The large model provides a supervision signal that allows RL training for the small model. Experimental evaluations
14 across five multi-hop QA datasets demonstrate that Collab-RAG substantially outperforms the baselines.

**Questions To Authors:**

1. It could be possible that the decomposed questions are wrong, but the final exact match is good. Could this be a problem for the RL training of the white-box LLM? If so, how to avoid it?
2. It is interesting that `w/o Accuracy Reward` can achieve comparable performance to Collab-RAG. I am wondering why Format Reward is a good single reward in this case. It is very likely the Format Reward is positive but the final answer is wrong, isn't it?

**Reasons To Accept:**

1. The proposed method shows strong improvement over baselines, including those using much larger LLMs, on all 5 benchmark datasets.
2. The proposed method can help relieve the need of expensive human annotations or resource-intensive model distillation.

**Reasons To Reject:**

1. While efficient, the approach still relies on potentially costly black-box LLMs for feedback
2. The preference optimization process, while powerful, adds training complexity and may require careful tuning to avoid overfitting. In addition, the generalization ability of a model trained on one dataset is not evaluated on other datasets.

---

> ### Author Response · Authors · 2025-06-02
> **Initial Rebuttal to Reviewer YbMU**
>
> We appreciate the reviewer’s careful reading of our paper and valuable suggestions. Our responses to the comments are provided below.
>
> ***
> > W1: While efficient, the approach still relies on potentially costly black-box LLMs for feedback
>
> A: Thank you for the thoughtful feedback. We would first like to clarify that during training, we only use GPT-4o-mini for creating feedback signals, which costs less than `$10` for model training. That said, it is possible that the reader model could be replaced with a white-box language models (e.g. Qwen-2.5-14b Instruct), potentially enabling greater parameter efficiency. While this is beyond the scope of our current study, we view it as a valuable avenue for future work and will include this discussion in the revised version of the paper.
>
> ***
> > W2: The preference optimization process, while powerful, adds training complexity and may require careful tuning to avoid overfitting. In addition, the generalization ability of a model trained on one dataset is not evaluated on other datasets.
>
> A: Thank you for the comment. Regarding **training complexity**, we would like to emphasize that our method does not rely on extensive hyperparameter tuning. Most hyperparameters are kept fixed, as detailed in Section 5.2. Additionally, we include a hyperparameter sensitivity analysis in Table 6 (Appendix C.1), which shows that our model remains relatively stable across different values of $\beta$ and $k$. For instance, the standard deviation in performance across six hyperparameter configurations is only 0.4.
>
> For the **generalization ability** of our model, we point out that StrategyQA and Bamboogle are indeed out-of-domain datasets that are not used during training. The results in Table 1 indicate that Collab-RAG consistently outperforms baseline approaches on both benchmarks, which verifies its effectiveness in generalizing to unseen data.
>
> Besides, recent work [1] also suggests that preference optimization is generally more robust to overfitting than SFT, and often exhibits stronger generalization.
>
> >Reference:
> >
> >[1] Kirk, Robert, et al. "Understanding the effects of rlhf on llm generalisation and diversity." ICLR 2024
>
> ***
> > Q1: It could be possible that the decomposed questions are wrong, but the final exact match is good. Could this be a problem for the RL training of the white-box LLM? If so, how to avoid it?
>
> A: Thanks for raising this question! We agree that in principle, there may be cases where incorrect decompositions still lead to correct final answers, which could potentially introduce noise during reinforcement learning. However, in practice, we did not observe this issue to be significant. This is likely because the reader model we use (GPT-4o-mini) is relatively weak; when decompositions are poor, it typically fails to recover correct answers.
>
> To empirically validate this, we conducted a human evaluation of decomposition quality using a 3-point scale:
>
> - 0: Low quality (irrelevant decomposition)
>
> - 1: Medium quality (partially useful but incomplete or noisy)
>
> - 2: High quality (relevant, and complete decomposition)
>
> We randomly sampled 20 examples per dataset and report the average decomposition scores below:
>
> | Model     | Average Score |
> |------|------|
> | Frozen Qwen-2.5-3B     | 1.27  |
> | Frozen LLaMA-3.1-8B    | 1.32  |
> | Frozen GPT-4o-mini     | 1.43   |
> | Frozen GPT-4o     | 1.61   |
> | Collab-RAG 3B     | 1.60   |
> | Collab-RAG 8B    | 1.69   |
>
> These results demonstrate that our trained models achieve decomposition quality on par with GPT-4o (for the 3B variant) and exceed all baselines with the 8B variant, providing evidence that our optimization is effective and produces high-quality decompositions.
>
> Looking ahead, one way to further mitigate potential reward misalignment would be to incorporate hybrid reward signals, e.g., leveraging a separate LLM judge to evaluate both the correctness of the final answer and the quality of the decomposition. This could ensure that high rewards are only assigned when both elements align with the intended reasoning structure.
>
> ***
> > Q2: It is interesting that w/o Accuracy Reward can achieve comparable performance to Collab-RAG. I am wondering why Format Reward is a good single reward in this case. It is very likely the Format Reward is positive but the final answer is wrong, isn't it?
>
> A:  Thank you for pointing this out, and we apologize for the confusion. To clarify, the “w/o Accuracy Reward” setting removes only the *accuracy* component from the Accuracy Reward, while retaining the *exact match (EM)* component as a correctness signal. It does not mean we rely solely on the Format Reward or remove all answer-based supervision. We will revise the text to make this clearer and avoid potential misunderstanding in future versions.
>
> ***
> Thank you once again for your insightful review. We appreciate your feedback on our work. Feel free let us know if you have any further questions, and we are happy to discuss further.

---

### Official Review · Reviewer_vqF1 · 2025-05-19

**Rating:** 6
**Confidence:** 4
**Ethics Flag:** 1

**Summary:**

This paper introduces a method to improve multi-hop question answering (QA) by facilitating interaction between a white-box small language model (SLM) and a black-box large language model (LLM). In this setup, the SLM is responsible for query decomposition and improved retrieval, while the black-box LLM acts as the reader and synthesizer to produce the final answer. To train the query decomposer, the authors propose a self-improving training strategy that leverages feedback from the black-box LLM. This creates a multi-turn interaction loop, which iteratively refines the decomposition strategy and ultimately improves answer generation. The approach is evaluated on five multi-hop QA datasets, showing an average performance gain of 6.6% across exact match (EM), accuracy, and F1-score metrics.

**Reasons To Accept:**

The paper demonstrates promising results through a comprehensive experimental setup, benchmarking against a broad range of strong baselines.

**Reasons To Reject:**

While the proposed approach in interesting, the paper requires additional evidence to justify its necessity:

unsupported efficiency claims: The claim regarding the "efficiency" of Collab-RAG in improving reasoning and retrieval is not well-supported. Given that the method involves an iterative interaction between models, it is intuitively less efficient than single-step QA approaches. The paper does not clearly define what metric is used to measure "efficiency" and provides no quantitative comparison to existing methods in this regard.

impact of retrieval: although the results suggest that the method is retriever-agnostic, it remains unclear whether performance gains are from improved retrieval (because of query decomposition) or from the simplification of complex queries in the generation phase.

Limited evaluation metrics: The evaluation relies primarily on EM and F1-score. While standard, these metrics are limited in capturing the quality of generated answers. Also, "accuracy" is not clearly defined. Human judgment or an automated eval should be included.

---

> ### Author Response · Authors · 2025-06-02
> **Initial Rebuttal to Reviewer vqF1 (Part 2)**
>
> > W3: Limited evaluation metrics: The evaluation relies primarily on EM and F1-score. While standard, these metrics are limited in capturing the quality of generated answers. Also, "accuracy" is not clearly defined. Human judgment or an automated eval should be included.
>
> A: Thanks for this suggestion. We follow prior works [1,2,3] to use “accuracy” as a binary measure indicating whether the predicted answer contains the ground-truth answer. Per your suggestion, we incorporate automated evaluation using GPT-4.1 as an LLM judge. We compare Collab-RAG against strong baselines using the same reader model (GPT-4o), and report the LLM-judged answer correctness across four datasets:
>
> | Method           | HotpotQA | MusiQue | 2WikiMultiHopQA | Bamboogle |
> |------------------|----------|---------|------------------|-----------|
> | IRCOT            | 73.8     | 50.4    | 79.2             | 68.8      |
> | Iter-RetGen      | 73.2     | 44.2    | 76.2             | 63.2      |
> | RAG-Star         | 75.0     | 42.4    | 75.8             | --        |
> | Collab-RAG w/ Qwen-3B     | 79.0     | 50.2    | 82.4             | 76.0      |
> | Collab-RAG w/  LLaMA-8B    | 79.2     | 54.0    | 82.0             | 79.2      |
>
> As the results show, Collab-RAG consistently outperforms these baselines under LLM-based evaluation. We will incorporate the evaluation results in the revised version of the paper.
>
>
> >References:
> >
> >[1] Asai et al. "Self-rag: Learning to retrieve, generate, and critique through self-reflection." ICLR 2024.
> >
> >[2] Mallen et al. "When Not to Trust Language Models: Investigating Effectiveness of Parametric and Non-Parametric Memories." ACL 2023.
> >
> >[3] Jeong et al. "Adaptive-RAG: Learning to Adapt Retrieval-Augmented Large Language Models through Question Complexity." NAACL 2024.
>
> ***
>
> Thank you again for your review. We hope our response could address your concerns. If you have any further questions, we would be happy to discuss them further.

---

> ### Author Response · Authors · 2025-06-02
> **Initial Rebuttal to Reviewer vqF1 (Part 1)**
>
> We are grateful for the reviewer’s constructive remarks and suggestions. We respond to each point in the following.
>
> ***
> > W1: unsupported efficiency claims: The claim regarding the "efficiency" of Collab-RAG in improving reasoning and retrieval is not well-supported. Given that the method involves an iterative interaction between models, it is intuitively less efficient than single-step QA approaches. The paper does not clearly define what metric is used to measure "efficiency" and provides no quantitative comparison to existing methods in this regard.
>
> A: Thank you for the feedback! We clarify that the “efficiency” of Collab-RAG actually means the **parameter efficiency**, as Collab-RAG-3B outperforms a frozen 32B LLM in question decomposition (10.7× larger) and Collab-RAG-8B surpasses a frozen 72B LLM (9× larger).
>
> Regarding **time efficiency during inference**, we acknowledge that Collab-RAG adopts an iterative reasoning and retrieval framework. However, this is also the case for most recent strong baselines (e.g., IRCoT, Search-o1, IterDRAG, RAG-Star, RQ-RAG, Iter-RetGen, RAG-Gym), which similarly rely on multi-step retrieval and decomposition. To make our claim more concrete, we present a comparison of inference time and average performance (EM) over QA datasets across representative methods using GPT-4o-mini as the reader model:
>
> | Method           | Inference Time per Query | Avg QA Performance (EM) |
> |------------------|--------------------------|--------------------------|
> | Standard RAG     | 0.6s                     | 28.4                     |
> | Search-o1        | >10s                     | 43.9                    |
> | IRCoT  (few shot)          | 2.9s                     | 52.5                     |
> | Collab-RAG w/ Qwen-2.5-3B    | 2.0s                     | 53.8                     |
> | Collab-RAG w/ Llama-3.1-8B    | 2.3s                     | 55.4                     |
>
> While Collab-RAG introduces more overhead than standard one-shot RAG, it delivers over 20% absolute EM improvement, which we believe justifies the additional computation.
> Collab-RAG is more time-efficient than Search-o1, which incurs significantly higher latency due to generations of thinking traces. Compared to IRCOT, Collab-RAG yields better accuracy with faster inference, all while using a smaller backbone model—further justifying its practical efficiency.
>
> We will revise the manuscript to clarify our use of “efficiency” mainly refers to “parameter efficiency”, and include this quantitative comparison for transparency.
>
> ***
> > W2: impact of retrieval: although the results suggest that the method is retriever-agnostic, it remains unclear whether performance gains are from improved retrieval (because of query decomposition) or from the simplification of complex queries in the generation phase.
>
> A: Thank you for the insightful question. We believe that both enhanced retrieval and simplifying complex queries through decomposition can contribute to improved model performance. To investigate their individual contributions, we conducted two additional ablation studies:
>
> 1. **Subquestion-only reasoning**: The reader model is asked to answer each decomposed sub-question *without any retrieved context*, then combine the intermediate answers to derive the final answer.
>
> 2. **Subquestion-based retrieval without decomposition**: The reader is prompted with the original question, augmented by a set of top-5 passages retrieved using the decomposed sub-questions, but *without explicitly decomposing the question in the prompt*.
>
> | Method                                      | HotpotQA Acc | HotpotQA EM | HotpotQA F1 | MusiQue Acc | MusiQue EM | MusiQue F1 |
> |--------------------------------------------|--------------|-------------|-------------|-------------|------------|------------|
> | RAG                                         | 58.4         | 41.8        | 57.1        | 22.6        | 11.4       | 23.4       |
> | Subquestion-only reasoning  | 60.6         | 44.4        | 60.3        | 32.2        | 16.6       | 33.1       |
> | RAG w/ passage retrieved using decomposed questions  | 64.0         | 47.8        | 62.5        | 34.8        | 19.8       | 34.4       |
> | Collab-RAG                                  | 66.2         | 53.0        | 65.6        | 45.8        | 26.4       | 42.4       |
>
> The results show that both variants outperform standard single-turn RAG but still fall short of our full Collab-RAG approach. This supports the importance of both high-quality retrieval and explicit decomposition.

---

> ### Comment · Reviewer_vqF1 · 2025-06-09
> **Updated score**
>
> Thank you for addressing my comments. The paper would benefit from a more in-depth analysis of the section discussing the impact of retrieval. However, if the authors incorporate the feedback provided in these reviews, the paper becomes convincing overall. I have updated my score to 6.

---

> > ### Author Response · Authors · 2025-06-11
> >
> > Thank you for raising the score! We'll include the additional experiments and analysis in our next version.

---

### Official Review · Reviewer_sLb3 · 2025-05-23

**Rating:** 7
**Confidence:** 5
**Ethics Flag:** 1

**Summary:**

The paper presents Collab-RAG, a framework for retrieval-augmented generation that combines a trainable query decomposer (SLM) with a black-box LLM reader. The decomposer generates sub-questions, which are used to retrieve context and generate answers via the black-box LLM. The SLM is trained using DPO, guided by rewards derived from the final answer quality.

**Questions To Authors:**

See above.

**Reasons To Accept:**

* It is interesting to formulate query decomposition as a learnable task using a white-box SLM.
* The empirical evaluation is comprehensive, spanning multiple datasets, LLM readers, and decomposer model sizes.

**Reasons To Reject:**

* I find that the core motivation of this work (improving complex question answering via iterative retrieval and reasoning) largely overlaps with that of IRCoT. However, unlike IRCoT’s simple and training-free zero-shot prompting approach, Collab-RAG introduces notable system complexity and computational overhead by requiring iterative training of a white-box SLM using DPO. Also, the improvement in EM scores is sometimes marginal, which raises concerns about whether such added complexity is practically justified in real-world deployments.


* I am concerned about the fairness of the comparison to prior black-box RAG methods, particularly IRCoT. Specifically, IRCoT relies on GPT-3 as the reader, while Collab-RAG uses GPT-4o-mini, a significantly more capable model. Given the substantial performance gap between GPT-3 and GPT-4 series models in reasoning and contextual understanding, the observed performance improvements may be partially attributed to the underlying LLM upgrade rather than the proposed collaborative framework alone. To more clearly isolate the contribution of Collab-RAG’s decomposition and optimization mechanisms, I encourage the authors to include a comparison where IRCoT-style retrieval and reasoning is applied using the same GPT-4o-mini as the backbone.


* Related to the question above, the authors categorize IRCoT and related methods as “reference only” in Table 1, yet provide no clear explanation for this distinction. Given that IRCoT is conceptually aligned with Collab-RAG in both motivation and design and that its implementation is publicly available and easily runnable with the same black-box LLM used in this work (e.g., GPT-4o-mini), this categorization appears arbitrary. I encourage the authors to clarify their rationale for this labeling and to report the results under a unified experimental setup. In other words, for a fair and meaningful comparison, I strongly recommend rerunning IRCoT (and other black-box-based baselines) using the same retriever and LLM configuration as Collab-RAG, and reporting the full set of metrics accordingly.


* The number and granularity of sub-questions can critically affect both effectiveness and efficiency. It would be valuable to include such analyses.


* As the final answer accuracy is used as a proxy for sub-question quality during training, I think it would be better to include the analyses on the direct assessment of the quality of the decompositions themselves.

---

> ### Author Response · Authors · 2025-06-02
> **Initial Rebuttal to Reviewer sLb3 (Part 3)**
>
> > W4: The number and granularity of sub-questions can critically affect both effectiveness and efficiency. It would be valuable to include such analyses.
>
> A: Thank you for the suggestion. In response, we analyzed sub-question distributions on HotpotQA and MusiQue. The table below shows both the proportion of different decomposition lengths and the corresponding average EM when using GPT-4o-mini as the reader:
>
> | # of Subquestions  | 2      | 3      | 4      | 5      | >5     |
> |--------------------|--------|--------|--------|--------|--------|
> | **HotpotQA**       |        |        |        |        |        |
> | Llama-3.1-8B Proportion for Numbers of Subquestions | 15.8%  | 37.2%  | 22.6%  | 17.4%  | 7.0%   |
> | Llama-3.1-8B Average EM (with GPT-4o-mini as the reader) | 46.7%  | 50.5%  | 37.2%  | 44.8%  | 37.2%  |
> | Ours Finetuned Model Proportion for Numbers of Subquestions  | 34.6%  | 43.2%  | 16.0%  | 5.0%   | 1.2%   |
> | Ours Finetuned Model Average EM (with GPT-4o-mini as the reader)  | 49.6%  | 55.3%  | 46.3%  | 55.2%  | 47.3%  |
> | **MusiQue**        |        |        |        |        |        |
> | Llama-3.1-8B Proportion for Numbers of Subquestions | 21.6%  | 33.4%  | 23.0%  | 14.8%  | 7.2%   |
> | Llama-3.1-8B Average EM  (with GPT-4o-mini as the reader) | 39.7%  | 20.3%  | 9.1%   | 9.9%   | 4.8%   |
> | Ours Finetuned Model Proportion for Numbers of Subquestions | 41.8%  | 36.0%  | 16.2%  | 5.2%   | 0.8%   |
> | Ours Finetuned Model Average EM (with GPT-4o-mini as the reader) | 40.2%  | 20.8%  | 10.0%  | 10.4%  | 6.7%   |
>
>
> These results suggest that **Collab-RAG offers improved granularity control**. Our finetuned model generates fewer but more targeted sub-questions, as indicated by the increased proportion of cases with 2–3 sub-questions. This likely reduces unnecessary decomposition overhead.  Additionally, across all granularities, our decompositions lead to consistently higher downstream EM, demonstrating their effectiveness in guiding the reader. We will include this analysis in the revised manuscript.
>
> ***
> > W5: As the final answer accuracy is used as a proxy for sub-question quality during training, I think it would be better to include the analyses on the direct assessment of the quality of the decompositions themselves.
>
> A:  Thank you for the suggestion. In response, we conducted a human evaluation to directly assess the quality of the generated sub-questions. We employed a three-point scale:
> - 0: Low quality (irrelevant decomposition)
> - 1: Medium quality (partially useful but incomplete or noisy)
> - 2: High quality (relevant, and complete decomposition)
> And randomly select 20 questions from each dataset for this human evaluation. The average results are shown in the following table:
>
> | Model                  | Average Score |
> |------------------------|---------------|
> | Frozen Qwen-2.5-3B     | 1.27          |
> | Frozen LLaMA-3.1-8B    | 1.32          |
> | Frozen GPT-4o-mini     | 1.43          |
> | Frozen GPT-4o          | 1.61          |
> | Collab-RAG 3B          | 1.60          |
> | Collab-RAG 8B          | **1.69**          |
>
> These results show that Collab-RAG-3B already matches GPT-4o's quality with significantly fewer parameters, and the 8B variant surpasses all baselines, demonstrating the effectiveness of our decomposition strategy.
>
> ***
>
> Thank you once again for your review. We appreciate your feedback on our work. If you have any further questions, please let us know. We would be happy to discuss them further.

---

> > ### Comment · Reviewer_sLb3 · 2025-06-06
> > **Thanks for your response!**
> >
> > Thanks for your very detailed responses, and I hope these points are included in the updated version.
> > Therefore, I will raise my score.

---

> > > ### Author Response · Authors · 2025-06-07
> > >
> > > Thanks for reading our rebuttal! We are glad that our responses have addressed your concerns. We will make sure to include those points in our next version.

---

> ### Author Response · Authors · 2025-06-02
> **Initial Rebuttal to Reviewer sLb3 (Part 2)**
>
> > [Continued Rebuttal for W2 and W3]:
>
> That said, in response to your feedback, we have re-implemented and evaluated representative black-box RAG methods, including IRCoT (zero-shot and 10-shot), BlendFilter, and IterDRAG, under the same settings as Collab-RAG, using both GPT-4o-mini and GPT-4o as the reader LLMs. The results are presented below:
>
> | Baseline                     | StrategyQA |        |        |  |    HotpotQA    |       |  |     MusiQue   |       |   |    2WikiMQA   |       |  |  Bamboogle     |       | Avg. EM       |
> |-----------------------------|------------|--------|--------|----------|--------|-------|---------|--------|-------|-----------|-------|-------|------------|-------|-------|----------------|
> |                             | EM         |        |        | Acc      | EM     | F1    | Acc     | EM     | F1    | Acc       | EM    | F1    | Acc        | EM    | F1    |                |
> | **GPT-4o-mini as reader LLM** |            |        |        |          |        |       |         |        |       |           |       |       |            |       |       |                |
> | IRCoT-zero shot              | 77.7       |        |        | 58.8     | 44.8   | 57.6  | 31.4    | 18.8   | 31.2  | 54.8      | 43.6  | 56.6  | 37.6       | 28.8  | 41.6  | 42.7           |
> | IRCoT-10 shot              | 79.0       |        |        | 62.4     | 49.6   | 61.8  | 42.4    | 25.0   | 40.5  | 73.6      | 59.4  | 70.3  | 53.2       | 49.6  | 61.8  | 52.5           |
> | BlendFilter                 | 75.9       |        |        | 59.6     | 45.6   | 59.9  | 24.8    | 12.8   | 29.3  | 56.2      | 44.6  | 55.7  | 60.8       | 52.8  | 61.2  | 46.3           |
> | IterDRAG                    | 78.6       |        |        | 60.6     | 46.8   | 59.8  | 27.8    | 19.6   | 28.1  | 74.2      | 63.6  | 71.7  | 54.0       | 46.4  | 60.7  | 51.0           |
> | **Collab-RAG w/ Qwen-2.5-3B**  | 82.0       |        |        | 67.2     | 51.6   | 66.2  | 41.6    | 25.4   | 39.6  | 79.4      | 63.0  | 74.5  | 55.4       | 47.2  | 62.0  | **53.8 (+2.5%)** |
> | **Collab-RAG w/ Llama-3.1-8B** | 81.6       |        |        | 66.2     | 53.0   | 65.6  | 45.8    | 26.4   | 42.4  | 79.0      | 63.2  | 74.6  | 59.4       | 52.8  | 64.8  | **55.4 (+5.5%)** |
> | **GPT-4o as reader LLM**    |            |        |        |          |        |       |         |        |       |           |       |       |            |       |       |                |
> | IRCoT-zero shot              | 78.6       |        |        | 64.2     | 48.0   | 63.7  | 33.8    | 22.4   | 33.5  | 61.4      | 51.4  | 61.0  | 60.8       | 46.4  | 56.9  | 49.4           |
> | IRCoT-10 shot              | 81.2       |        |        | 66.4     | 52.8   | 66.0  | 44.2    | 29.4   | 43.9  | 78.0      | 62.2  | 72.9  | 66.4       | 57.6  | 70.0  | 56.6           |
> | BlendFilter                 | 78.6       |        |        | 64.4     | 51.6   | 64.7  | 36.2    | 19.6   | 36.5  | 77.2      | 59.6  | 72.4  | 67.2       | 59.2  | 70.4  | 53.7           |
> | IterDRAG                    | 81.2       |        |        | 64.2     | 51.2   | 65.1  | 38.8    | 25.0   | 37.9  | 78.8      | 68.4  | 77.2  | 62.0       | 55.2  | 67.8  | 56.2           |
> | **Collab-RAG w/ Qwen-2.5-3B**  | 82.5       |        |        | 68.6     | 55.6   | 68.3  | 43.6    | 26.2   | 40.0  | 82.0      | 67.0  | 77.9  | 66.4       | 60.0  | 70.6  | **58.3 (+3.0%)** |
> | **Collab-RAG w/ Llama-3.1-8B** | 82.9       |        |        | 69.2     | 54.4   | 68.3  | 47.2    | 29.0   | 43.4  | 81.0      | 67.2  | 77.0  | 69.6       | 63.2  | 74.0  | **59.3 (+4.8%)** |
>
> As shown, Collab-RAG consistently outperforms these baselines by 2.5–5.5% when using GPT-4o-mini and by 3.0–4.8% with GPT-4o, while also incurring lower API costs – e.g., IRCoT zero-shot and 10-shot settings require 1.4x and 2.5x more API usage, respectively. These improvements highlight that our gains stem not just from stronger LLMs but from the effectiveness of the collaborative training framework. We will incorporate these updated results and clarify label annotations in the revised paper.

---

> ### Author Response · Authors · 2025-06-02
> **Initial Rebuttal to Reviewer sLb3 (Part 1)**
>
> Thank you for your insightful comments and for taking the time to review our paper. We respond to your suggestions as follows.
>
> ***
> > W1: I find that the core motivation of this work (improving complex question answering via iterative retrieval and reasoning) largely overlaps with that of IRCoT. However, unlike IRCoT’s simple and training-free zero-shot prompting approach, Collab-RAG introduces notable system complexity and computational overhead by requiring iterative training of a white-box SLM using DPO. Also, the improvement in EM scores is sometimes marginal, which raises concerns about whether such added complexity is practically justified in real-world deployments.
>
> A: Thank you for raising this important point. We agree that Collab-RAG introduces additional complexity during the training phase due to the iterative optimization of a white-box SLM using DPO. However, we would like to emphasize that this training cost is *incurred only once*. After training, the model can be deployed for inference across various datasets without additional training.
>
> In contrast, IRCoT relies on repeated black-box prompting during inference, which becomes increasingly expensive at scale. For instance, when using the same GPT-4o backbone, IRCoT incurs **1.4x API costs than Collab-RAG** and this additional cost becomes **2.5x when few-shot in-context (passage, answer) pairs are provided in the context**. Despite this, Collab-RAG still outperforms IRCoT with the same backbone (please refer to the responses in W2/W3 for detailed results).
>
> While prompting-based methods are suitable for small-scale data, Collab-RAG provides a more scalable and cost-effective solution for large-scale deployments. We'll clarify this trade-off and practical implication in the revision.
>
> ***
> > W2: I am concerned about the fairness of the comparison to prior black-box RAG methods, particularly IRCoT. Specifically, IRCoT relies on GPT-3 as the reader, while Collab-RAG uses GPT-4o-mini, a significantly more capable model. Given the substantial performance gap between GPT-3 and GPT-4 series models in reasoning and contextual understanding, the observed performance improvements may be partially attributed to the underlying LLM upgrade rather than the proposed collaborative framework alone. To more clearly isolate the contribution of Collab-RAG’s decomposition and optimization mechanisms, I encourage the authors to include a comparison where IRCoT-style retrieval and reasoning is applied using the same GPT-4o-mini as the backbone.
>
> > W3: Related to the question above, the authors categorize IRCoT and related methods as “reference only” in Table 1, yet provide no clear explanation for this distinction. Given that IRCoT is conceptually aligned with Collab-RAG in both motivation and design and that its implementation is publicly available and easily runnable with the same black-box LLM used in this work (e.g., GPT-4o-mini), this categorization appears arbitrary. I encourage the authors to clarify their rationale for this labeling and to report the results under a unified experimental setup. In other words, for a fair and meaningful comparison, I strongly recommend rerunning IRCoT (and other black-box-based baselines) using the same retriever and LLM configuration as Collab-RAG, and reporting the full set of metrics accordingly.
>
> A:  Thank you for the thoughtful suggestion. Regarding the “reference only” label in Table 1, we would like to clarify that these baselines are included for reference due to several factors:
>
> - For methods using **white-box LLMs**, while these models typically have significantly fewer parameters than black-box LLMs, these models are often trained with additional training data (e.g. RankRAG and ChatQA use large-scale (>100k) SFT data for finetuning), making direct comparison less meaningful. We mainly demonstrate that our method can achieve good performance with relatively fewer training examples.
>
> - For **black-box LLM-based baselines**, such as IRCoT and BlendFilter, they often rely on powerful but expensive models for decomposition, whereas our method targets more efficient setups.
>
> - Additionally, some baselines (e.g., BlendFilter, IterDRAG) do not have **open-source implementations**, making it challenging to fully replicate their exact experimental conditions.
>
> As a result, the numbers for these methods are drawn from their original papers and are marked as “reference only.” We also would like to highlight that for many other baselines (e.g. Iter-RetGen, RQ-RAG, RAG-Star, RAFE etc.), we use the same backbone and training data to **ensure the comparison is fair**.
>
> *[Due to space limit, the remaining response to this question is in the next post]*

---

### Official Review · Reviewer_j6Ne · 2025-05-25

**Rating:** 7
**Confidence:** 4
**Ethics Flag:** 1

**Summary:**

The paper focuses on the task of answering complex questions using retrieved passages and a reader language model. The proposed method called "Collab-RAG" consists of a two step pipelined approach: (i) in the first step, a smaller language model decomposes a complex question into a series of simple (or factoid) atomic questions such that these atomic questions can be answered using a collection of relevant passages; (ii) In the second stage, these sub-questions and the relevant passages are given as input to a larger black-box LLM that processes them to generate the final answer.

In Collab-RAG, the trainable module is the smaller language model. The proposed training approach consists of first fine-tuning it using rejection sampling followed by iterative DPO (Direct Preference Optimization). Experiments are conducted on multiple multi-hop QA datasets and demonstrate comparable or new state-of-the-art results against recent baselines. The paper also contains multiple ablation studies to demonstrate the usefulness of different design decisions.

**Questions To Authors:**

* To my knowledge, the paper leverages a static dense retriever. It will be good to see a discussion in the paper when the retriever is further finetuned on the retrieval task for each of the datasets.

* I would suggest adding this baseline to the paper: Given a complex question, retrieve all the relevant documents such as top-500, top-1000, top-5000, etc. from the retriever and give as input to the reader (large) language model to answer the question. It will be useful to understand how much the reader LM can use the long-context to understand the answer the question.

* As Collab-RAG is an iterative approach during training, it will be nice to briefly include a discussion of the system time requirement of the system in the paper during training and inference and how it compares to current approaches.

**Reasons To Accept:**

* The paper is very well-written. I really enjoyed reading the paper. Almost everything is quite well done. The proposed method is elegant and carefully explained. The experiments are well conceived, and the results are compared against a lot of baselines.

* The idea of finetuning a language model to breakdown a complex questions into a simpler one is not necessarily novel, the main novelty consists in the training algorithm. The training approach first trains the query decomposer language model on high-quality sub-questions dataset and then further finetunes it using preference optimization using feedback as a relevance signal from a larger language model.

* The experimental methodology look sound and the results are compared against many recent approaches and strong baselines. Using their proposed training method, authors are able to get good empirical performance gains that either matches or outperforms the previous approaches.

**Reasons To Reject:**

* Regarding experiments with different question decomposition language models: the paper has mainly used a couple of smaller language models for finetuning. I would like to see a discussion when black box models such as Gemini, GPT-4o, etc. are used for question decomposition.

* Specificity of proposed method: Is this approach only valid for complex questions. Can the author demonstrate the effect of this framework on simple question answering tasks as well. Some of the datasets that can be used for evaluation are SimpleQA, Natural Questions etc.

* Finetuning the reader models: Almost all of the results in the paper are based on using larger reader models as black-box. It will be interesting to see how the effect of finetuning readers models as well and this impacts the iterative DPO training.

---

> ### Author Response · Authors · 2025-06-02
> **Initial Rebuttal to Reviewer j6Ne (Part 2)**
>
> > Q2: I would suggest adding this baseline to the paper: Given a complex question, retrieve all the relevant documents such as top-500, top-1000, top-5000, etc. from the retriever and give as input to the reader (large) language model to answer the question. It will be useful to understand how much the reader LM can use the long-context to understand the answer the question.
>
> A: Thank you for the suggestion. Per your suggestion, we conducted additional experiments with larger $k$ values (specifically $k=50$ and $k=100$) to further explore the limits of long-context reading by large language models.
>
> | Model           | Dataset        | Acc  | EM   | F1   |
> |----------------|----------------|------|------|------|
> | GPT-4o-mini     | HotpotQA (k=50)       | 56.6 | 40.4 | 55.6 |
> |                | HotpotQA (k=100) | 55.8 | 38.8 | 54.2 |
> |                | MusiQue (k=50)        | 23.4 | 12.0 | 23.9 |
> |                | MusiQue (k=100)  | 22.0 | 10.8 | 22.9 |
> | GPT-4o          | HotpotQA  (k=50)      | 63.0 | 45.2 | 62.2 |
> |                | HotpotQA (k=100) | 63.8 | 45.6 | 62.9 |
> |                | MusiQue   (k=50)      | 27.6 | 16.4 | 28.3 |
> |                | MusiQue (k=100)  | 27.2 | 15.8 | 27.9 |
>
>
> Due to context length and budget constraints (e.g., inference with k=1,000 Wikipedia passages consumes ~128k tokens and costs over `$150` per dataset), we were unable to evaluate larger values such as $k$=500, 1000, 5000. Nonetheless, the results clearly show diminishing returns—and even degradation—as $k$ increases.
>
> These findings suggest that simply increasing the number of retrieved documents can lead to information overload and confusion, thereby reducing answer quality. We will include this discussion and additional results in the revised version of the paper.
>
> ***
> > Q3: As Collab-RAG is an iterative approach during training, it will be nice to briefly include a discussion of the system time requirement of the system in the paper during training and inference and how it compares to current approaches.
>
> A:  Thanks for the comment. We list the time used in training (for both SFT and iDPO) and inference in the following table:
>
> | Stage                        | 3B       | 8B       |
> |-----------------------------|----------|----------|
> | SFT Training                | 20 mins  | 30 mins  |
> | iDPO Sampling per iteration | 1.5 h    | 2.3 h    |
> | iDPO Training per iteration | 15 mins  | 25 mins  |
> | Inference time per query    | 2.0 s    | 2.3 s    |
>
> We would like to highlight that although our method introduces some additional training overhead due to its iterative optimization, the **inference efficiency remains comparable** to existing multi-hop RAG approaches (e.g., RQ-RAG, Iter-RetGen, IRCOT), which similarly involve multi-step question decomposition, answering, and passage retrieval.
>
> ***
> Thank you again for your review. We hope our response could address your concerns. If you have any further questions, we would be happy to discuss them further.

---

> > ### Comment · Reviewer_j6Ne · 2025-06-10
> > **Thanks for your response**
> >
> > Dear Authors,
> > Thanks for putting the efforts on crafting the rebuttal. I am overall satisfied with the response.
> > I would like to keep my score unchanged.

---

> ### Author Response · Authors · 2025-06-02
> **Initial Rebuttal to Reviewer j6Ne (Part 1)**
>
> Thank you so much for your positive and encouraging feedback. We are glad to hear that you enjoyed reading the paper! Below, we address your comments and suggestions in details:
> ***
> > W1: Regarding experiments with different question decomposition language models: the paper has mainly used a couple of smaller language models for finetuning. I would like to see a discussion when black box models such as Gemini, GPT-4o, etc. are used for question decomposition.
>
> A: Thank you for the suggestion. We have already included baselines that use black-box LLMs (GPT-4o and GPT-4o-mini) for question decomposition, as shown in **Table 1** (“RAG w/ GPT-4o-mini Decompose (Khot et al.)” and “RAG w/ GPT-4o Decompose (Khot et al.)”). As the results indicate, Collab-RAG continues to outperform these baselines despite relying on smaller backbone models with 3-8B parameters. This highlights the effectiveness and parameter efficiency of our approach, even when compared to powerful proprietary models.
>
> ***
> > W2: Specificity of proposed method: Is this approach only valid for complex questions. Can the author demonstrate the effect of this framework on simple question answering tasks as well. Some of the datasets that can be used for evaluation are SimpleQA, Natural Questions etc.
>
> A: Thank you for the question. Our method is tailored for **complex questions** that require multi-step reasoning and benefit from decomposition into simpler sub-questions.  In contrast, simple questions, such as those in datasets like SimpleQA or Natural Questions, are generally self-contained and do not require decomposition. As such, applying Collab-RAG to these cases would be unnecessary and unlikely to yield meaningful gains. We will clarify this intended scope and the specificity of our approach in the next version of the paper.
>
> ***
> > W3: Finetuning the reader models: Almost all of the results in the paper are based on using larger reader models as black-box. It will be interesting to see how the effect of finetuning readers models as well and this impacts the iterative DPO training.
>
> A: Thank you for the thoughtful suggestion. In this work, we opt to use black-box LLMs as reader models due to their strong zero-shot capabilities, which make them well-suited for handling the decomposed sub-questions produced by our framework. That said, we agree that replacing the reader with a fine-tuned SLM is a promising direction, potentially enabling greater parameter efficiency and further integration with the DPO-based training pipeline. While this is beyond the scope of our current study, we view it as a valuable avenue for future work and will include this discussion in the revised version of the paper.
>
> ***
> > Q1: To my knowledge, the paper leverages a static dense retriever. It will be good to see a discussion in the paper when the retriever is further finetuned on the retrieval task for each of the datasets.
>
> A: Thank you for the insightful suggestion. First, we would like to clarify that **Collab-RAG do not rely on a specific retriever**. As shown in Table 3, we have evaluated Collab-RAG with different off-the-shelf dense retrievers, including COCO-DR, E5-Large, and GTE. The consistent performance gains across these variants demonstrate the robustness of our approach to different retrieval configurations. While our current work uses static retrievers, we agree that fine-tuning retrievers on dataset-specific retrieval tasks is a promising direction. Since Collab-RAG is retriever-agnostic, we expect that **stronger, fine-tuned retrievers could further enhance performance**. We also believe that designing effective retriever fine-tuning strategies in multi-hop settings is an important open challenge that may merit a dedicated study. We will include this discussion in the revised version of the paper.

---

### Comment · Area_Chair_WrTH · 2025-06-08
**Please acknowledge author response (discussion period ends on June 10th)**

Dear reviewers,

Please acknowledge the author's response.
See if the response resolves the raised issue/questions, and  let the authors know how you think.
Lastly, please adjust your scores if necessary.

Best,
AC

---

### Decision · Program_Chairs · 2025-07-08

**Decision:**

Accept

**Comment:**

The paper presents Collab-RAG, a collaborative framework for multi-hop question answering (QA), where a small open-source language model (SLM) decomposes complex questions and a larger black-box language model (LLM) synthesizes final answers. The small model is trained via supervised fine-tuning followed by iterative Direct Preference Optimization (DPO) using feedback from the black-box model. Experiments on five multi-hop QA datasets show that the approach achieves competitive or superior results compared to strong baselines.

Reviewers generally agreed about the paper’s clarity, empirical strength, and novel training procedure, and that the proposed approach offers a promising alternative to knowledge distillation and helps reduce annotation overhead.

Reviewers have shown concerns regarding the ablation study, fair baseline comparisons, and the cost for training/inference; however, the authors have responded well with requested experiments and analyses.

I recommend that the authors reflect these new results and clarifications made in the rebuttal process in the final version of this paper, along with other reviewer suggestions, such as recommendations for presentation improvements and additional related work. **I especially think unifying the backbone reader (as reviewers SLb3, k4rg pointed out) and adding an ablation study would be important!**

Adding one more related work regarding utilizing larger models for training another in DPO manner:
Choi et al., Model-based Preference Optimization in Abstractive Summarization without Human Feedback, EMNLP 2024